# A conditional gene-based association framework integrating isoform-level eQTL data reveals new susceptibility genes for schizophrenia

Xiangyi Li[1,2,3†], Lin Jiang[4†], Chao Xue[1,2,3], Mulin Jun Li[5], Miaoxin Li[1,2,3,6*]

[1]Program in Bioinformatics, Zhongshan School of Medicine and The Fifth Affiliated Hospital, Sun Yat-sen University, Guangzhou, China; [2]Key Laboratory of Tropical Disease Control (Sun Yat-sen University), Ministry of Education, Guangzhou, China; [3]Center for Precision Medicine, Sun Yat-sen University, Guangzhou, China; [4]Research Center of Medical Sciences, Guangdong Provincial People's Hospital, Guangdong Academy of Medical Sciences, Guangzhou, China; [5]The Province and Ministry Co-sponsored Collaborative Innovation Center for Medical Epigenetics, Tianjin Medical University, Tianjin, China; [6]Guangdong Provincial Key Laboratory of Biomedical Imaging and Guangdong Provincial Engineering Research Center of Molecular Imaging, The Fifth Affiliated Hospital, Sun Yat-sen University, Zhuhai, China

**\*For correspondence:**
limiaoxin@mail.sysu.edu.cn

†These authors contributed equally to this work

**Competing interest:** The authors declare that no competing interests exist.

**Abstract** Linkage disequilibrium and disease-associated variants in the non-coding regions make it difficult to distinguish the truly associated genes from the redundantly associated genes for complex diseases. In this study, we proposed a new conditional gene-based framework called eDESE that leveraged an improved effective chi-squared statistic to control the type I error rates and remove the redundant associations. eDESE initially performed the association analysis by mapping variants to genes according to their physical distance. We further demonstrated that the isoform-level eQTLs could be more powerful than the gene-level eQTLs in the association analysis using a simulation study. Then the eQTL-guided strategies, that is, mapping variants to genes according to their gene/isoform-level variant-gene *cis*-eQTLs associations, were also integrated with eDESE. We then applied eDESE to predict the potential susceptibility genes of schizophrenia and found that the potential susceptibility genes were enriched with many neuronal or synaptic signaling-related terms in the Gene Ontology knowledgebase and antipsychotics-gene interaction terms in the drug-gene interaction database (DGIdb). More importantly, seven potential susceptibility genes identified by eDESE were the target genes of multiple antipsychotics in DrugBank. Comparing the potential susceptibility genes identified by eDESE and other benchmark approaches (i.e., MAGMA and S-PrediXcan) implied that strategy based on the isoform-level eQTLs could be an important supplement for the other two strategies (physical distance and gene-level eQTLs). We have implemented eDESE in our integrative platform KGGSEE (http://pmglab.top/kggsee/#/) and hope that eDESE can facilitate the prediction of candidate susceptibility genes and isoforms for complex diseases in a multi-tissue context.

## Editor's evaluation

This manuscript describes a new method of identifying disease-relevant risk genes from genome wide association studies by using a conditional gene-based method (named eDESE). The authors apply this new approach in an analysis of schizophrenia datasets, and identify meaningful biological

processes and potential drug repurposing candidates. Thus, this new method could provide improved gene prioritization for fine-mapping and functional studies of specific diseases.

## Introduction

Genome-wide association studies (GWASs) have been used to identify genotype-phenotype associations for over a decade, and thousands of single-nucleotide polymorphisms (SNPs) have been revealed for their associations with hundreds or thousands of complex human diseases (*Visscher et al., 2017*; *Gallagher and Chen-Plotkin, 2018*). Nevertheless, conventional GWASs analyses have limited power to produce a complete set of susceptibility variants for complex diseases (*Tam et al., 2019*). Because most susceptibility SNPs only have small effects on a complex phenotype, conventional SNP-based association tests are generally underpowered to reveal susceptibility variants after multiple-testing corrections. Moreover, the susceptibility variants scattering randomly throughout the genome are often in strong linkage disequilibrium (LD) with numerous neutral SNPs, which makes the discrimination of truly causal variants from GWAS hits quite difficult (*Tam et al., 2019*). Finally, more than 90% of the disease-associated variants are in non-coding regions of the genome, and many of them are far from the nearest known gene, and it remains a challenge to link genes and a complex phenotype through the non-coding variants (*Schaub et al., 2012*; *Maurano et al., 2012*). Accordingly, corresponding methodological strategies have been proposed to make up, at least partly, for the issues mentioned above.

First, gene-based approaches can reduce the multiple-testing burdens by considering the association between a phenotype and all variants within a gene (*Neale and Sham, 2004*). Assigning a variant to a gene according to the physical distance of the variant from gene boundary is one of the most popular strategies for gene-based approaches. For example, MAGMA (Multi-marker Analysis of GenoMic Annotation), one of the most popular gene-based approaches, uses a multiple regression approach to incorporate LD between markers and detect multi-markers effects to perform gene-based analysis (*de Leeuw et al., 2015*). VEGAS, a versatile gene-based test for GWAS, incorporates information from a full set of markers (or a defined subset) within a gene and accounts for LD between markers by simulations from the multivariate normal distribution (*Liu et al., 2010*). GATES, a rapid gene-based association test that uses an extended Simes procedure to assess the statistical significance of gene-level associations (*Li et al., 2011*). SuSiE (sum of single effects), a novel and popular approach to variable selection in linear regression, can use summary statistics and LD to produce gene-level evidence of association in terms of Bayes Factor (*Wang et al., 2020*).

Second, evaluating the gene-phenotype associations at one gene conditioning on other genes can isolate true susceptibility genes from the redundant non-susceptibility genes (*Li et al., 2019*). *Yang et al., 2012* proposed an approximate conditional and joint association analysis method based on linear regression analysis to estimate the individual causal variant with GWAS summary statistics. Our previously proposed conditional gene-based association approach based on effective chi-squared statistics (ECS) could remove redundantly associated genes based on the GWAS p-values of variants. Comparing the conditional gene-based association approach with ECS, MAGMA, and VEGAS suggested that the former might be more powerful to predict biologically sensible susceptibility genes (*Li et al., 2019*).

Third, the observation that variants in the non-coding regions were enriched in the transcriptional regulatory regions implied that these variants might affect the disease risk by altering the genetic regulation of target genes (*Gallagher and Chen-Plotkin, 2018*). Integration of expression quantitative trait loci (eQTL) studies and GWAS has been used to investigate the genetic regulatory effects on complex diseases. As many complex diseases manifested themselves in certain tissues, using the eQTLs of potentially phenotype-associated tissues might help identify the true susceptibility genes in the tissue context (*Hekselman and Yeger-Lotem, 2020*). Based on the framework of MAGMA, a method called eMAGMA integrated genetic and transcriptomic information (e.g. eQTLs) in a tissue-specific analysis to identify risk genes and was applied to identify novel genes underlying the major depression disorder (*Gerring et al., 2021*; *Gerring et al., 2019*). S-PrediXcan was developed for imputing the genetically regulated gene expression component based on GWAS summary statistics and transcriptome prediction models built from the eQTL/splicing QTL dataset of the Genotype Tissue Expression (GTEx) project (*Barbeira et al., 2018*). Researchers have applied S-PrediXcan to

study genetic mechanisms of multiple complex traits (*Gamazon et al., 2019*; *Gamazon et al., 2018*; *Huckins et al., 2019*). In contrast to the considerable research focusing on integrating gene-level eQTLs with GWAS summary statistics, little attention has been paid to integrating isoform-level eQTLs with GWAS summary statistics. Michael Gandal et al. estimated the candidate risk genes of three psychiatric disorders based on GWAS summary statistics and isoform-level expression profiles. They emphasized the importance of isoform-level gene regulatory mechanisms in defining cell type and disease specificity (*Gandal et al., 2018*), and similar analyses and conclusions were generated for Alzheimer's disease (*Fan et al., 2021*).

Although much achievement has been attained, identifying independently phenotype-associated genes with high reliability remains challenging, especially for complex diseases. Conventional gene-based approaches mainly focus on variants close (say ±5 kilo base pairs) to genes boundary and omit the distal but important variant-gene associations. Gene-based association approaches using eQTLs to identify candidate susceptibility genes and isoforms might raise the utilization rate of the distal but important variant-gene associations. The present study aimed to build a more powerful conditional gene-based framework based on a new ECS and mainly guided by eQTLs. We also investigated whether isoform-level eQTLs in the phenotype-associated tissues can help predict more significant susceptibility genes than gene-level eQTLs. The main assumption is that isoform-level eQTLs may reflect the more real regulatory relationship than gene-level eQTLs. Thus, using isoform-level eQTLs can help predict novel susceptibility genes and isoforms that the conventional gene-based approaches and gene-level eQTLs strategy cannot find. The following is the formation procedure of the assumption: Gene-level and isoform-level eQTLs are predicted based on the gene-level and isoform-level (or transcript-level) expression profiles, and the gene-level expression profiles are computed by averaging the expression of multiple isoforms produced by the gene. Thus, gene-level expression profiles

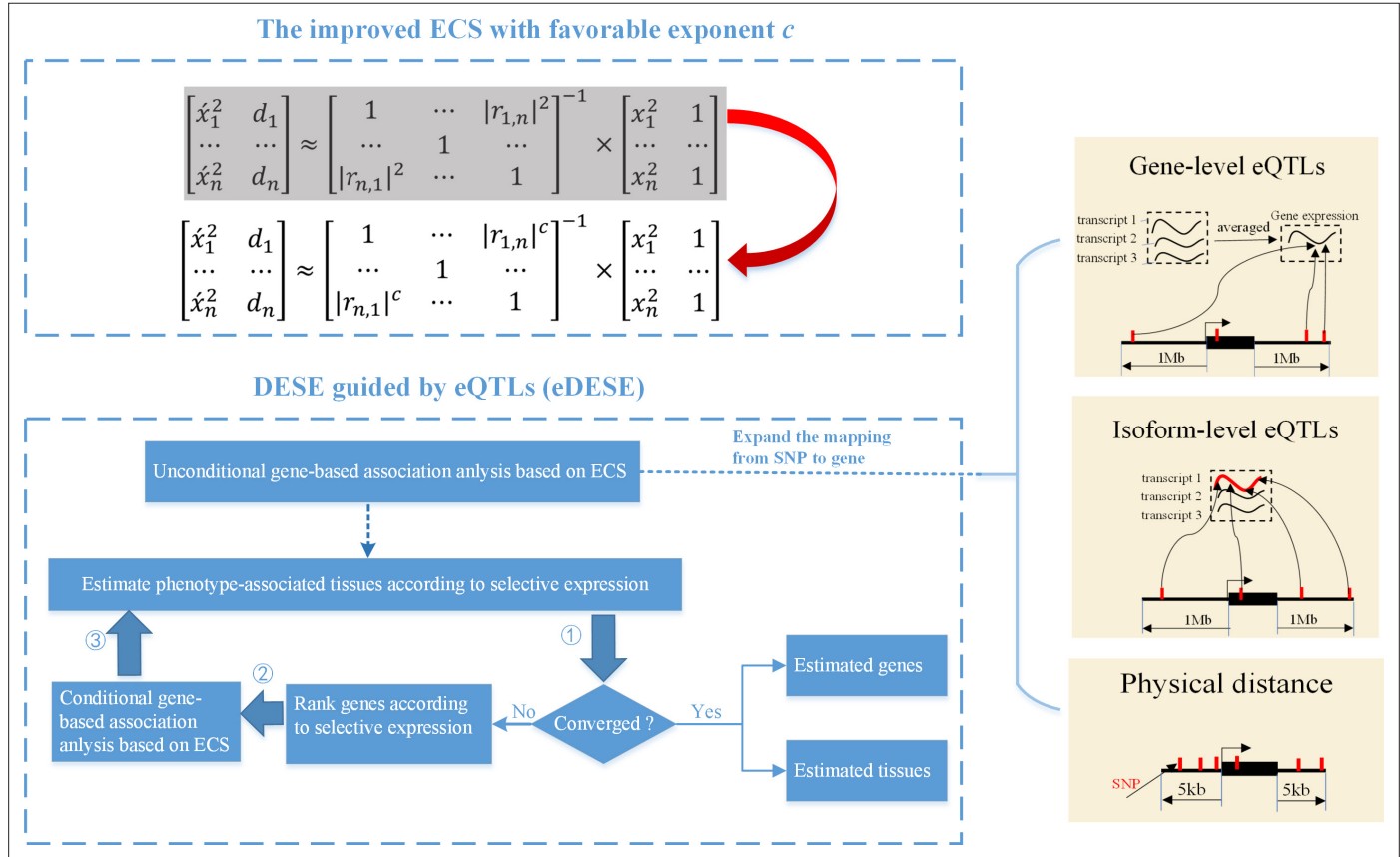

**Figure 1.** The advantages of eDESE over ECS and DESE. First, we proposed a new ECS and chose the best exponent $c$ between 1 and 2 to properly control the type I error. Second, we first adopted three strategies to map SNP to genes to perform the unconditional gene-based association analysis with the improved ECS. Then the unconditional gene-based association analysis results were put into the conditional gene-based association analysis and the following iterative procedure. 5 kb: 5000 base pairs. 1 Mb: $10^6$ base pairs.

may omit the expression heterogeneity among these isoforms, neutralize the opposite effects, and lower the power of gene-level eQTLs.

## Results

### Overview of the present study

We previously proposed an <u>e</u>ffective <u>c</u>hi-squared <u>s</u>tatistic called ECS for the unconditional and conditional gene-based association analysis (*Li et al., 2019*). Then we built a unified framework called DESE to estimate the potentially phenotype-associated tissues based on the conditional gene-based association analysis with ECS and gene selective expression analysis (*Jiang et al., 2019*). However, we found that the previous ECS was hindered by a potential type I error inflation issue and further undermined the accuracy of DESE. Here we proposed a new conditional gene-based association framework, eDESE (<u>e</u>QTL-guided <u>DESE</u>), which could also perform conditional gene-based association analysis and geneselective expression analysis, to systematically explore the susceptibility genes and tissues associated with complex diseases by using the GWAS summary statistics and multiple gene-variant mapping strategies. eDESE inherited the framework of DESE but had two important advantages over DESE. First, eDESE is built based on a new ECS, with which the type I error could be controlled within a proper level. Second, eDESE expands the conditional gene-based association analysis of DESE by not only using physically nearby SNPs but also using the gene-level and isoform-level *cis*-eQTLs associations (*Figure 1*).

To evaluate the performance of the new ECS and eDESE, we performed extensive simulations and a real data application to schizophrenia. Specifically, we organized the present study in several sequential parts that cover the optimizing the exponent of chi-squared statistics to control the type I error rates, applying the new ECS to perform conditional gene-based association analysis in simulation data and real-world schizophrenia GWAS summary statistics data. For simplicity and clarity, the model integrating different mapping strategies, that is, physically nearby SNPs (<u>dist</u>ance), <u>gene</u>-level and <u>isoform</u>-level variant-gene *cis*-eQTLs associations, were named eDESE:dist, eDESE:gene and eDESE:isoform, respectively.

### Choose the favorable exponent *c* for the correlation matrix of chi-squared statistics to control the type I error rates

We found that the exponent *c* in the correlation matrix of chi-squared statistics could determine the deviation of the p-values produced by the ECS tests against the uniform distribution. As shown in *Figure 2*, the *c* = 1.0 led to deflated p-values while the *c* = 2.0 led to inflated p-values in the upper tail of the Q-Q plot against the uniform distribution. This pattern was independent of sample sizes, variant number and phenotype distribution (for binary or continuous traits) (*Figure 2*). The stable

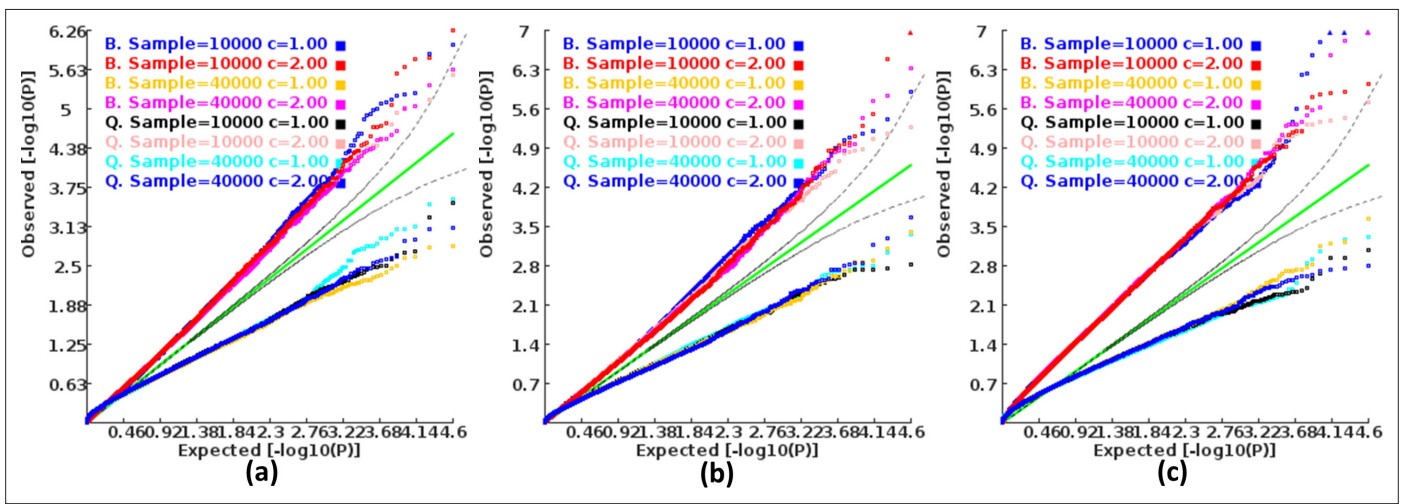

**Figure 2.** Q-Q plots of the p-value of the ECS test under null hypothesis based on the two extreme exponents (i.e. 1 and 2). (**a**), (**b**), and (**c**) represent the variant number of 50, 100, and 500, respectively.

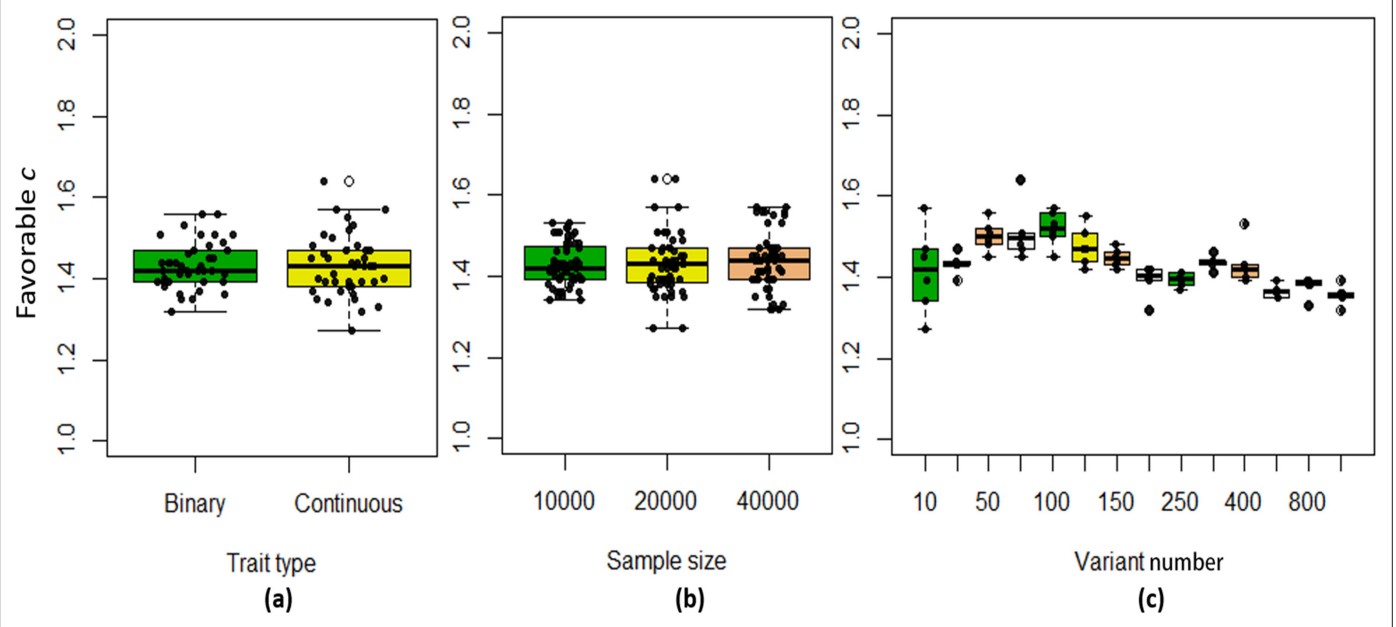

**Figure 3.** The boxplots of the favorable *c* values at different simulation scenarios. (**a**) Binary and continuous phenotypes; (**b**) Different sample sizes; (**c**) Different variant number.

trend determined by the *c* value also implied that the favorable *c*, which could properly control the type I error rate, measured by the minimal mean log fold change (MLFC), must be within range 1 and 2. Besides, our theoretical derivation also demonstrated that the *c* value should be within range 1 and 2. Moreover, it seemed that given the *c* value, the distributions of p-values were similar at different sample sizes and phenotype distributions. *Figure 3* shows the favorable *c* obtained by the grid search algorithm at 84 different scenarios. As shown in *Figure 3*, most majority p-values at the sample size 10,000 and 40,000 of binary or continuous phenotypes are overlapped. Again, the favorable *c* values were approximately independent of trait types, sample sizes and variant number. For the sake of simplicity, we proposed to use the averaged favorable *c* value, 1.432, and integrated it into the improved ECS.

## The type I error and power of the conditional gene-based association analysis based on the new effective chi-squared statistics (ECS)

We then investigated the type I error and power of the conditional gene-based association analysis based on the improved ECS with the favorable exponent *c* value. As shown in *Figure 4*, in six different scenarios, the conditional p-values of the genes without truly casual loci approximately follow the uniform distribution $U[0,1]$, regardless of the variance explained by its nearby genes. Moreover, the distribution of conditional p-value was similar to that produced by the conventional likelihood ratio test for the nested linear regression models. These results suggested that the conditional gene-based association analysis based on the improved ECS could produce valid p-values for statistical inference. In contrast, the unconditional association test based on the improved ECS produced an inflated p-value due to the indirect associations produced by the nearby causal genes in the LD block. Concerning the statistical power, we found that conditional gene-based association analysis based on the improved ECS produced smaller p-values than the likelihood ratio test (*Figure 5*), suggesting a higher statistical power of the former. Another advantage of conditional gene-based association analysis based on the improved ECS over the likelihood ratio test was that the former did not require individual genotypes. The reason might be that the degree of freedom in the latter was inflated by the LD among variants.

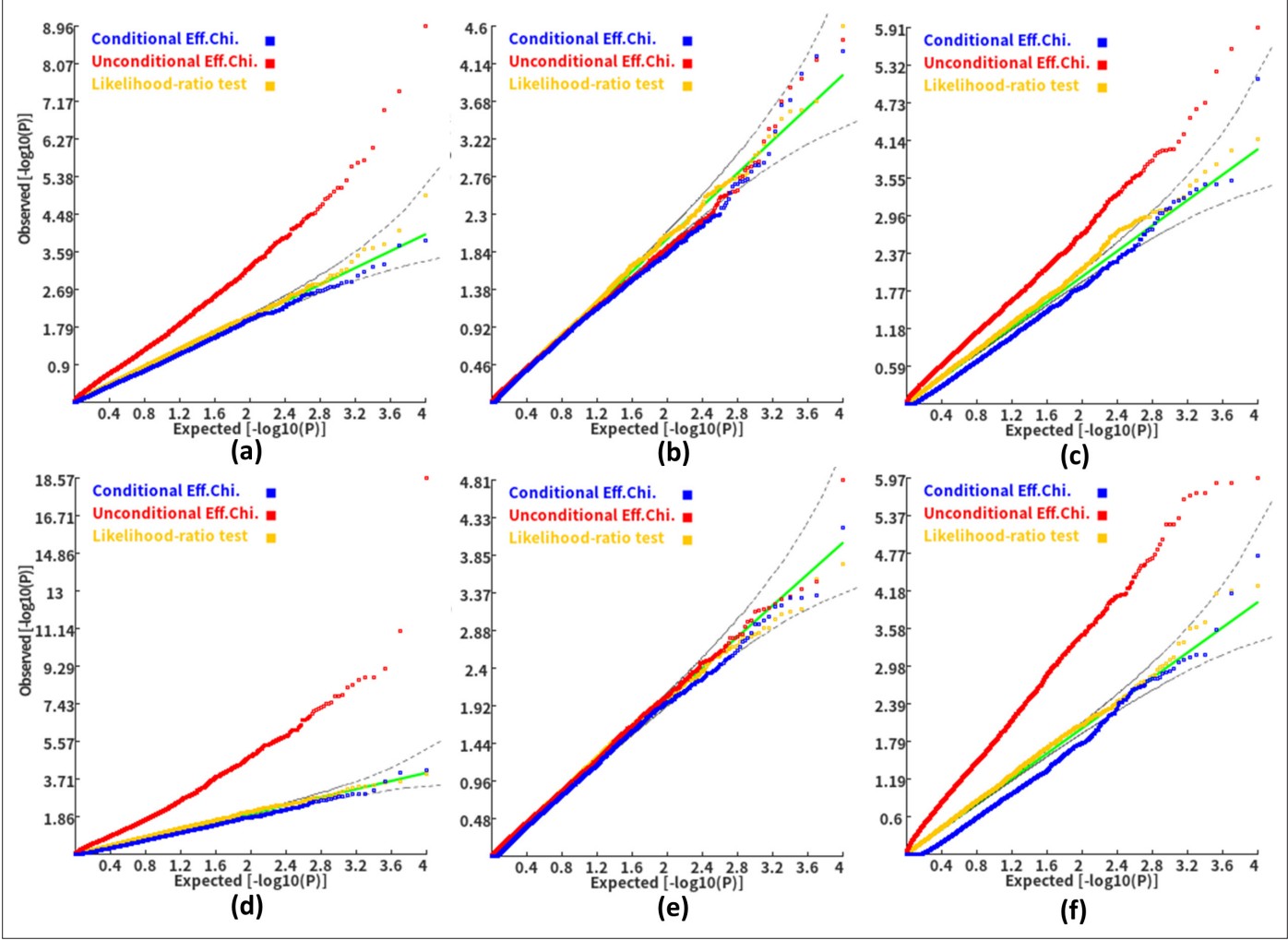

**Figure 4.** Q-Q plots of the conditional, unconditional gene-based association test and likelihood-ratio test under the null hypothesis. (**a**) and (**d**) two gene-variant pairs with the similar variant number (SIPA1L2 with 29 variants and LOC729336 with 30 variants). (**b**) and (**e**) two gene-variant pairs with the different variant number, and the first is larger than the second (CACHD1 with 41 variants and RAVER2 with eight variants). (**c**) and (**f**) Two gene-variant pairs with the different variant number, and the second is larger than the first (LOC647132 with five variants and FAM5C with 48 variants). (**a**), (**b**) and (**c**) The former gene has no QTL, and QTL explained 0.5 % of heritability in the latter gene. (**d**), (**e**) and (**f**) The former gene has no QTL, and QTL explained 1 % of heritability in the latter gene. Ten thousand phenotype datasets were simulated for each scenario. Unconditional Eff. Chi. (the red) represents unconditional association analysis at the former gene by the improved ECS. Conditional Eff. Chi (the blue) represents conditional association analysis at the former gene conditioning on the latter gene by the improved ECS. The likelihood ratio test (the yellow) was conducted based on the nested linear regression models.

## Apply eDESE:dist to predict the potential susceptibility genes of schizrenia

We had demonstrated that the conditional gene-based analysis based on the improved ECS was more powerful than the likelihood ratio test in each simulation scenario. To evaluate the performance of the ECS and eDESE in the real-world data, we used a recent large-scale GWAS summary statistics dataset (*Trubetskoy et al., 2022*) and gene expression profiles (GTEx v8) of multiple human tissues (*Consortium, 2020*) to identify the susceptibility genes of schizophrenia. We found that the improved ECS identified 739 significant genes without conditioning on gene-expression profiles. Furthermore, we also found 205 significant genes out of the above 739 genes identified by eDESE:dist based on the improved ECS by conditioning on the gene-level expression profiles (see details in *Supplementary file 1a*).

We then compared the significant susceptibility genes identified by eDESE:dist with that of MAGMA. We identified 619 significant susceptibility genes based on MAGMA. The significant gene

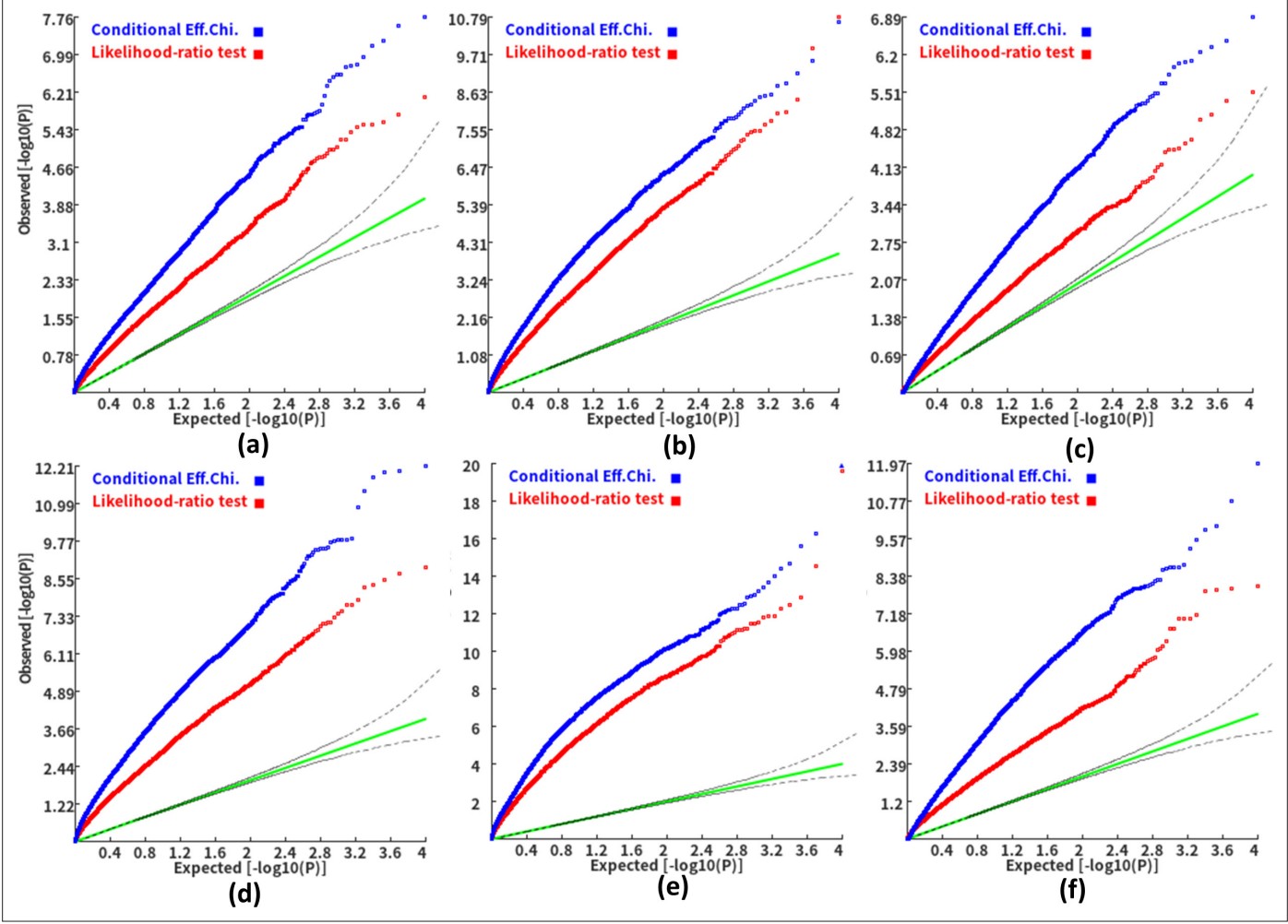

**Figure 5.** Q-Q plots of the conditional gene-based association test and likelihood-ratio test at different representative gene-variant pairs. The variant number of the two gene-variant pairs involved in (**a**)-(**f**) are the same as that in *Figure 4* legend. The difference is: in (**a**)-(**c**), the QTL in either gene (former and latter) explained 0.25% of heritability. In (**d**)-(**f**), the QTL in either gene explained 0.5% of heritability. One thousand phenotype datasets were simulated for each scenario.

count of MAGMA was about three times larger than that of eDESE:dist, which might partly result from the conditional gene-based analysis's advantage of removing the redundantly associated genes. Besides, more than half of the significant susceptibility genes identified by eDESE:dist were also identified by MAGMA (*Figure 6a*).

Further, we performed Gene Ontology (GO) enrichment analysis to study the functional annotations of these significant genes. Interestingly, we found that most GO:BP and GO:CC enrichment terms of the overlapped genes were neuronal-, dendrite- or synaptic signaling-related terms. The GO:MF enrichment terms of the overlapped genes were all about signaling transduction (see examples in *Figure 6b* and details in *Supplementary file 1b*). We then found that the unique genes identified by eDESE:dist were enriched with three GO:CC terms, that is, dendrite, dendritic tree and distal axon, which were all dendrite-related terms. However, although the unique genes identified by MAGMA were enriched with thirty-one GO terms, none of these terms were neuronal-, dendrite-, or synaptic signaling-related terms. Moreover, systematic text-mining results in PubMed showed that 67 of the 205 (~32.7%) and 170 of the 619 (~27.5%) potential susceptibility genes had at least two search hits for eDESE:dist and MAGMA, respectively (see details in *Supplementary file 1c*). The GO enrichment results and the text-mining results both implied the utility of eDESE:dist.

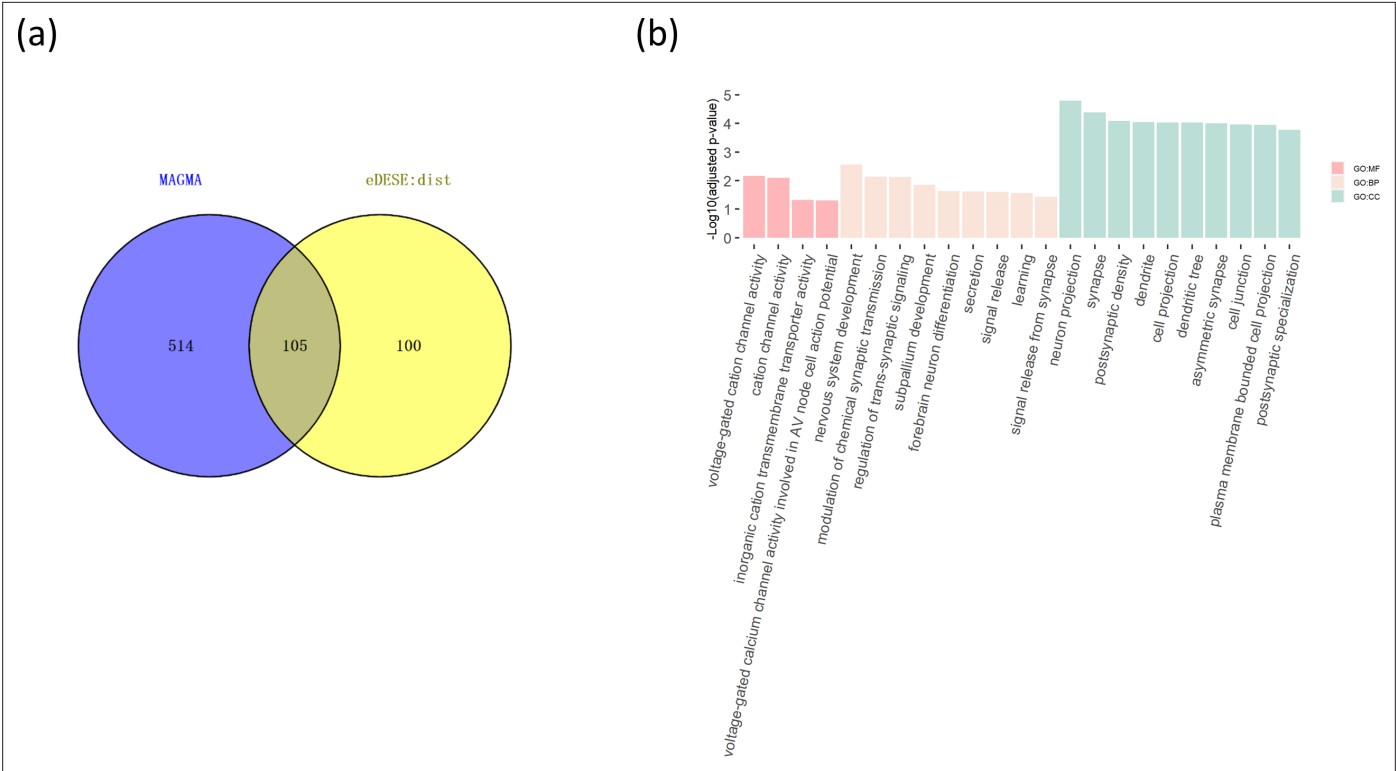

**Figure 6.** The comparison of the potential susceptibility genes for schizophrenia identified by MAGMA and eDESE:dist. (**a**) The Venn diagram shows the overlapped and unique genes identified by MAGMA and eDESE:dist. (**b**) The bar plot shows the top GO enrichment terms of the overlapped genes. MF: Molecular Function of GO. BP: Biological Process terms of GO. CC: Cellular Component terms of GO. The x-axis label represents the top ( ≤ 10) significant GO enrichment terms (MF, BP, and CC). The y-axis label represents the negative log10 of the adjusted p-value of each term. See also *Figure 6—source data 1*, *Figure 6—source data 2* and *Figure 6—source data 3*.

The online version of this article includes the following figure supplement(s) for figure 6:

**Source data 1.** Significant genes identified by MAGMA.

**Source data 2.** Significant genes identified by eDESE:dist.

**Source data 3.** Enrichment results of the common 105 genes identified by MAGMA and eDESE:dist.

## Evaluate the type I error and power of gene-level eQTLs and isoform-level eQTLs in association analysis based on ECS

Next, we mapped variants to genes (or isoforms) according to their variant-gene/isoform eQTLs associations. Since the isoform-level and corresponding gene-level expression profiles were quantified based on the same RNA-sequencing data, we then investigated the type I error and power of the association analysis by ECS based on the gene-level and isoform-level eQTLs.

We considered the multiple different scenarios that variants affected phenotype through regulating the gene expression. We simulated genotype data, gene-level and isoform-level expression data and corresponding phenotype data. In *Table 1*, AllVar means that all variants are used in the gene-based association analysis based on ECS. IsoeQTL denotes that the eQTL is associated with a susceptibility isoform. GeneQTL denotes that the eQTL is associated with a gene whose expression is averaged from three susceptibility isoforms (homogeneity). Gen3eQTL and Gen6eQTL denote that the eQTL is associated with a gene whose expression is averaged by three and six isoforms, one of which is the susceptibility isoform (heterogeneity), respectively.

As shown in *Table 1*, our simulations' type I error rates are controlled within 0% ~ 0.3% (on average <0.1%) according to the p-value threshold 0.001 (scenarios 1–3). As expected, in scenarios 4–6, where gene expression cannot affect the phenotype (Eg = 0 in *Table 1*), AllVar is much more powerful than eQTLs. Further, in scenarios 7–12, where gene expression can affect the phenotype, our results suggest that the powers of eQTLs roughly increase with the phenotype variance explained

**Table 1.** Type I error and power of different simulation scenarios in association analysis.

| Scenarios | Important parameters | | | Binary trait | | | | | Continuous trait | | | | |
|---|---|---|---|---|---|---|---|---|---|---|---|---|---|
| | Eg | Vgp | Vge | AllVar | IsoeQTL | GeneQTL | Gen3eQTL | Gen6eQTL | AllVar | IsoeQTL | GeneQTL | Gen3eQTL | Gen6eQTL |
| | | | | **Type I error** | | | | | | | | | |
| 1 | 0 | 0 | 0.05 | 0 | 0 | 0 | 0 | 0.002 | 0 | 0 | 0.001 | 0.002 | 0.003 |
| 2 | 0 | 0 | 0.15 | 0 | 0.002 | 0.001 | 0 | 0 | 0 | 0 | 0 | 0.001 | 0 |
| 3 | 0 | 0 | 0.3 | 0 | 0 | 0 | 0.002 | 0 | 0 | 0.001 | 0.001 | 0.002 | 0.002 |
| | | | | **Power** | | | | | | | | | |
| 4 | 0 | 0.005 | 0.05 | 0.251 | 0.036 | 0.022 | 0.034 | 0.031 | 0.246 | 0.032 | 0.019 | 0.032 | 0.038 |
| 5 | 0 | 0.005 | 0.15 | 0.219 | 0.021 | 0.013 | 0.023 | 0.032 | 0.301 | 0.025 | 0.017 | 0.037 | 0.043 |
| 6 | 0 | 0.005 | 0.3 | 0.229 | 0.028 | 0.017 | 0.021 | 0.034 | 0.282 | 0.024 | 0.017 | 0.025 | 0.039 |
| 7 | 0.1 | 0 | 0.05 | 0 | **0.017** | **0.019** | 0.006 | 0.001 | 0 | **0.017** | **0.027** | 0.009 | 0.001 |
| 8 | 0.1 | 0 | 0.15 | 0 | **0.213** | **0.221** | 0.113 | 0.054 | 0.002 | **0.245** | **0.245** | 0.132 | 0.068 |
| 9 | 0.1 | 0 | 0.3 | 0.018 | **0.704** | **0.659** | 0.581 | 0.388 | 0.027 | **0.72** | **0.686** | 0.607 | 0.446 |
| 10 | 0.1 | 0.005 | 0.05 | 0.288 | **0.052** | **0.076** | 0.043 | 0.043 | 0.313 | **0.063** | **0.091** | 0.05 | 0.041 |
| 11 | 0.1 | 0.005 | 0.15 | 0.403 | **0.33** | **0.302** | 0.199 | 0.134 | 0.46 | **0.357** | **0.334** | 0.229 | 0.136 |
| 12 | 0.1 | 0.005 | 0.3 | 0.569 | **0.778** | **0.738** | 0.677 | 0.485 | 0.62 | **0.805** | **0.774** | 0.712 | 0.512 |

Eg denotes the effect size of gene expression on phenotype. Vgp denotes phenotype variance explained by all variants. Vge denotes gene expression variance explained by all variants.

**Table 2.** The result about whether the brain was optimized as the schizophrenia-associated tissue based on each brain region's gene/isoform-level eQTLs.

| Brain regions | Gene-level eQTL | Isoform-level eQTL |
|---|---|---|
| **Brain-Anterior cingulate cortex (BA24)** | Yes | Yes |
| **Brain-Cerebellum** | Yes | Yes |
| **Brain-Frontal Cortex (BA9)** | Yes | Yes |
| **Brain-Hippocampus** | Yes | Yes |
| **Brain-Spinal cord (cervical c-1)** | Yes | Yes |
| Brain-Amygdala | Yes | No |
| Brain-Caudate (basal ganglia) | Yes | No |
| Brain-Cerebellar Hemisphere | Yes | No |
| Brain-Cortex | Yes | No |
| Brain-Hypothalamus | Yes | No |
| Brain-Nucleus accumbens (basal ganglia) | Yes | No |
| Brain-Putamen (basal ganglia) | No | No |
| Brain-Substantia nigra | Yes | No |

"Yes" denotes that brain (i.e., all thirteen brain tissues) was estimated as the significantly schizophrenia-associated tissue based on the gene/isoform-level eQTLs of the tissue. "No" denotes the contrary. The font names of the optimized brain regions are bold. See also Table 2—source data 1, Table 2—source data 2, Table 2—source data 3 and Table 2—source data 4.

The online version of this article includes the following source data for table 2:

**Source data 1.** Tissue significance estimated by eDESE:dist based on the gene-level expression profiles.

**Source data 2.** Tissue significance estimated by eDESE:dist based on the isoform-level expression profiles.

**Source data 3.** Tissue significance estimated by eDESE:gene based on the gene-level eQTLs of each brain region.

**Source data 4.** Tissue significance estimated by eDESE:isoform based on the isoform-level eQTLs of each brain region.

by all variants (Vgp in *Table 1*) and gene expression variance explained by all variants (Vge in *Table 1*). Moreover, in scenarios 7–12, associations test totally based on the susceptibility isoforms (IsoeQTL and GeneQTL in *Table 1*) are more powerful than those based on the gene-level eQTLs. While IsoeQTL is computed based on fewer (i.e. one) susceptibility isoforms than GeneQTL (i.e. three), the power of IsoeQTL is the equal of GeneQTL. Thus, our simulation results revealed that isoform-level eQTLs were more powerful than gene-level eQTLs in association analysis in the scenarios that gene expression could affect the phenotype.

## Estimate the potentially phenotype-associated tissues for schizophrenia

Like DESE, eDESE can produce phenotype-associated genes and tissues. Therefore, we firstly adopted eDESE:dist to predict the phenotype-associated tissues of schizophrenia and found that all thirteen brain regions were significantly associated with schizophrenia and ranked the top based on the gene-level and isoform-level expression profiles, respectively (see details in *Supplementary file 1d*).

Since all thirteen brain regions were predicted as the potentially phenotype-associated tissues of schizophrenia by eDESE:dist, removing the possible false positives would be necessary. Then we resorted to the eQTLs and assumed that if a tissue (say $T_1$) is a phenotype-associated tissue, potential susceptibility genes identified based on the eQTLs of $T_1$ will be more likely to be phenotype-associated genes and selectively express in $T_1$ or similar tissues. We then computed the gene-level and isoform-level eQTLs of all thirteen brain regions and predicted the potentially phenotype-associated tissues using eDESE:gene and eDESE:isoform, respectively. Our results showed that the brain (all thirteen brain regions as a whole) was predicted as the schizophrenia-associated tissue based on the

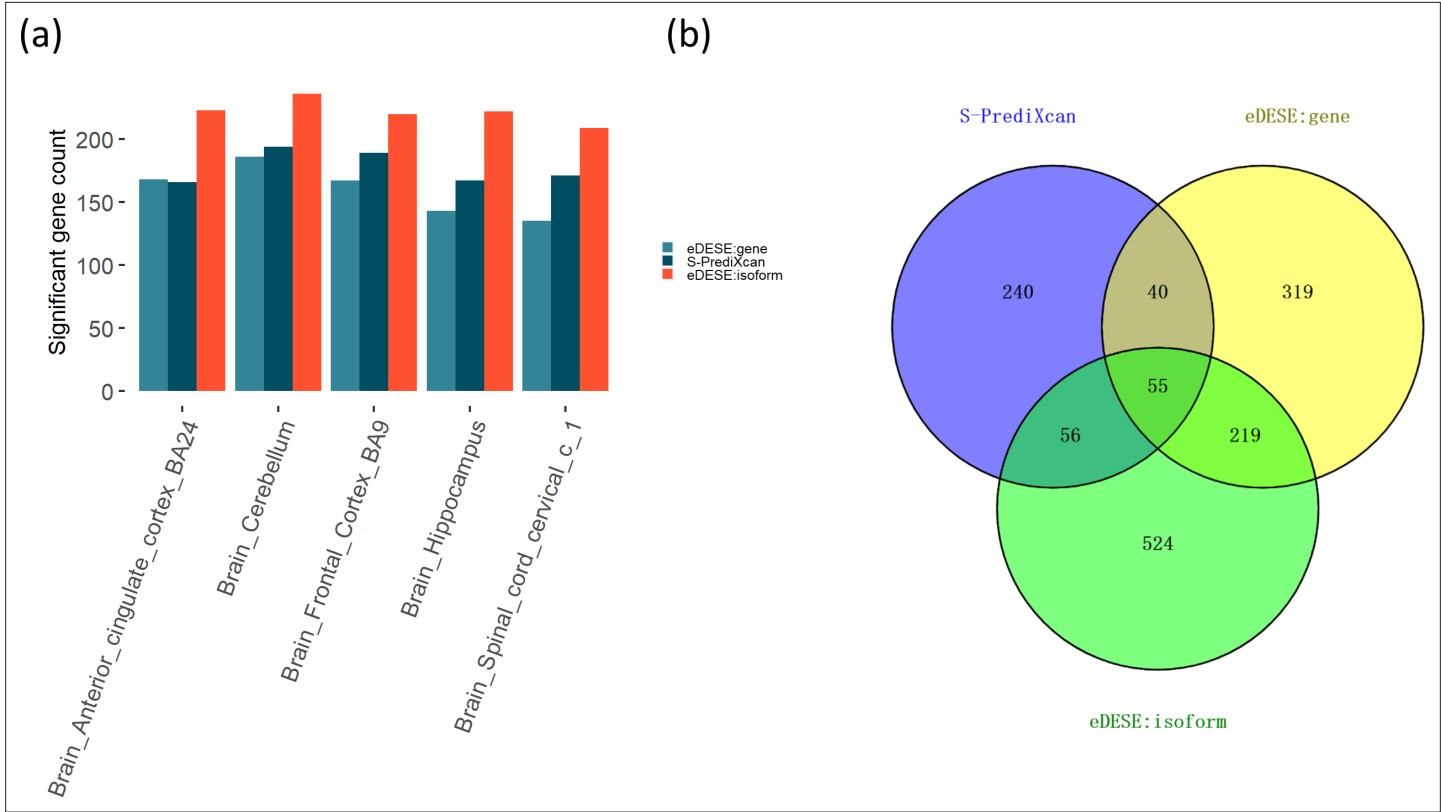

**Figure 7.** Comparison of the potential susceptibility genes identified by S-PrediXcan, eDESE:gene, and eDESE:isoform. (**a**) The bar plot shows the count of potential susceptibility genes in each of the five optimized brain regions. (**b**) The Venn diagram shows the count of the overlapped and unique genes identified by S-PrediXcan, eDESE:gene and eDESE:isoform in the five optimized brain regions. See also *Figure 7—source data 1* and *Figure 7—source data 2*.

The online version of this article includes the following figure supplement(s) for figure 7:

**Source data 1.** The count of potential susceptibility genes in each of the five optimized brain regions.

**Source data 2.** The potential susceptibility genes identified by S-PrediXcan, eDESE:gene and eDESE:isoform in the five optimized brain regions.

gene-level eQTLs of twelve brain regions, respectively (*Table 2*). However, on the more precise resolution, we found brain was predicted as schizophrenia-associated tissue only based on the isoform-level eQTLs of five brain regions, respectively (*Table 2*). Thus, the five brain regions were collectively optimized as the potentially phenotype-associated tissues of schizophrenia by eDESE:dist, eDESE:gene, and eDESE:isoform.

In contrast, we predicted schizophrenia-associated tissues based on the gene-level and isoform-level eQTLs of Muscle Skeletal and Skin Sun Exposed Lower Leg, whose sample sizes were bigger than all the other tissues in GTEx(v8). The prediction results of Muscle Skeletal and Skin Sun Exposed Lower Leg showed that neither of them was predicted to be significant tissues of schizophrenia by using their gene-level and isoform-level eQTLs (see details in *Supplementary file 1e*), which demonstrated, at least partly, our assumption. Thus, our results showed the utility of integrating the three models of eDESE for optimizing the phenotype-associated tissues.

## The comparison of eDESE:isoform versus eDESE:gene and S-PrediXcan

### eDESE:isoform can identify more potential susceptibility genes

Our simulation study demonstrated that association analysis based on the improved ECS with the isoform-level eQTLs was more powerful than with the gene-level eQTLs. We further tested this finding in the real data and identified the potential susceptibility genes for schizophrenia using the eDESE:isoform and eDESE:gene, respectively. We found the number of potential susceptibility genes identified by eDESE:isoform was larger than that of eDESE:gene and S-PrediXcan under the same adjustment

**Table 3.** The important examples of potential susceptibility genes exclusively predicted by eDESE:isoform.

| Gene name | # of hits in PubMed |
|---|---|
| RGS4 | > 100 |
| TCF4 | > 100 |
| RANGAP1 | > 100 |
| GRIA1 | 80 |
| GRM3 | 76 |
| TSPO | 39 |
| TPH2 | 35 |
| FEZ1 | 31 |
| ZDHHC8 | 24 |
| VRK2 | 23 |
| KCNN3 | 20 |
| NCAM1 | 20 |
| MIP | 15 |
| SLC39A8 | 14 |
| DLG1 | 14 |
| BDNF-AS | 13 |
| FGA | 13 |
| ADRA1A | 12 |
| MAPT | 10 |

The online version of this article includes the following source data for table 3:

**Source data 1.** The PubMed search hits of the unique potential susceptibility genes of schizophrenia identified by eDESE:isoform (compared with S-PrediXcan and eDESE:gene).

filter cutoff (*Figure 7a*, see details in *Supplementary file 1fg and h*). We further combined the potential susceptibility genes of the five optimized brain regions identified by S-PrediXcan, eDESE:gene and eDESE:isoform, respectively. Still, we found that the susceptibility genes exclusively identified by eDESE:isoform were the most among the three models (*Figure 7b*).

## Potential susceptibility genes identified by eDESE:isoform were supported by more published studies

Then we searched the PubMed database with the combined susceptibility gene set (i.e. 391 genes for S-PrediXcan, 633 genes for eDESE:gene, and 854 genes for eDESE:isoform, *Figure 7b*) of all five optimized brain regions. We found that 135 genes for S-PrediXcan, 170 genes for eDESE:gene and 247 genes for eDESE:isoform had at least one search hit which reported the associations of these genes with schizophrenia in PubMed database, respectively (see details in *Supplementary file 1ij and k*).

We also found 138 of the 524 (26.3%) potential susceptibility genes exclusively predicted by eDESE:isoform each had a least one search hit. Moreover, we found that 19 of the 524 (3.6%) potential susceptibility genes each had at least 10 supported papers in PubMed (*Table 3*, see details in ). Interestingly, *TCF4* (transcription factor 4), *RGS4* (regulator of G protein signaling 4) and *RANGAP1* (Ran GTPase activating protein 1) were reported by more than 100 papers. *RGS4* is reported to be biased expressed in brain (*O'Leary et al., 2016*). *TCF4* and *RANGAP1* are broadly expressed in the brain, and *TCF4* may play an important role in nervous system development (*O'Leary et al., 2016*).

Furthermore, we applied the Hetionet (v1.0) (*Himmelstein et al., 2017*), which encodes knowledge from millions of biomedical studies to connect diseases, genes, anatomies and more, to investigate the above top associations. We set the source node as the susceptibility gene and the target node as schizophrenia using the 'Connectivity search' function. We found that *RGS4* and *RANGAP1* both had multiple significant meta path types indicating their potential associations and mechanisms associated with schizophrenia.

## Potential susceptibility genes identified by eDESE:isoform were enriched with more biologically sensible GO terms

Next, we performed the GO enrichment analysis and found that the potential susceptibility genes identified by eDESE guided by the eQTLs (eDESE:gene and eDESE:isoform) had more biologically sensible GO enrichment terms than S-PrediXcan (*Table 4*). In addition, the GO enrichment results also showed that the potential susceptibility genes identified by eDESE:isoform were enriched with more neuronal, dendritic or synaptic signaling-related biological process GO terms than S-PrediXcan and eDESE:gene.

We further performed the GO enrichment analysis based on the combined gene lists of all five optimized brain regions for S-PrediXcan, eDESE:gene and eDESE:isoform. However, we found that

**Table 4.** The GO enrichment terms of the potential susceptibility genes in each optimized brain region identified by S-PrediXcan, eDESE:gene and eDESE:isoform.

| Tissue name | *S-PrediXcan | eDESE:gene | eDESE:isoform |
|---|---|---|---|
| Brain-Anterior cingulate cortex (BA24) | - | Regulation of gap junction assembly (BP) | Potassium ion transmembrane transporter activity (MF); potassium: chloride symporter activity (MF); **nervous system development (BP)**; **generation of neurons (BP)**; **neurogenesis (BP)** |
| Brain-Cerebellum | Ion binding (MF); cation binding (MF); metal ion binding (MF); intracellular organelle (CC) | Dendrite (CC); dendritic tree (CC); neuron projection (CC); postsynapse (CC); synapse (CC) | Voltage-gated calcium channel activity involved in cardiac muscle cell action potential (MF); **nervous system development (BP)** |
| Brain-Frontal Cortex (BA9) | Intracellular organelle (CC); intracellular membrane-bounded organelle (CC) | Somatodendritic compartment (CC); dendrite (CC); dendritic tree (CC); synaptic vesicle membrane (CC); exocytic vesicle membrane (CC) | - |
| Brain-Hippocampus | Ion binding (MF) | - | Postsynaptic density (CC); asymmetric synapse (CC) |
| Brain-Spinal cord (cervical c-1) | - | High voltage-gated calcium channel activity (MF); voltage-gated calcium channel activity involved in AV node cell action potential (MF); regulation of B cell tolerance induction (BP); positive regulation of B cell tolerance induction (BP); L-type voltage-gated calcium channel complex (CC) | Nitrogen compound transport (BP); organelle (CC) |

*MF: Molecular Function terms of GO. BP: Biological Process terms of GO. CC: Cellular Component terms of GO.

the 55 genes commonly identified by the three models were enriched with no GO term. Furthermore, the 240 unique genes identified by S-PrediXcan were also enriched with no GO term. On the other hand, the 319 unique genes identified by the eDESE:gene were enriched with 'integral component of synaptic vesicle membrane' (CC) and several general GO terms. In comparison, the 524 unique genes identified by eDESE:isoform were enriched with a considerable number of GO terms, in which a few neuronal, dendritic or synaptic signaling-related GO terms were found (*Supplementary file 1I*).

## Potential susceptibility genes identified by eDESE:isoform were significantly enriched in a biologically sensible consensus module in the brain weighted gene co-expression network

We then tested the enrichment of the potential susceptibility genes in the consensus modules of the brain weighted gene co-expression network. We found that the potential susceptibility genes identified by eDESE:gene and eDESE:isoform based on the gene-level and isoform-level eQTLs of

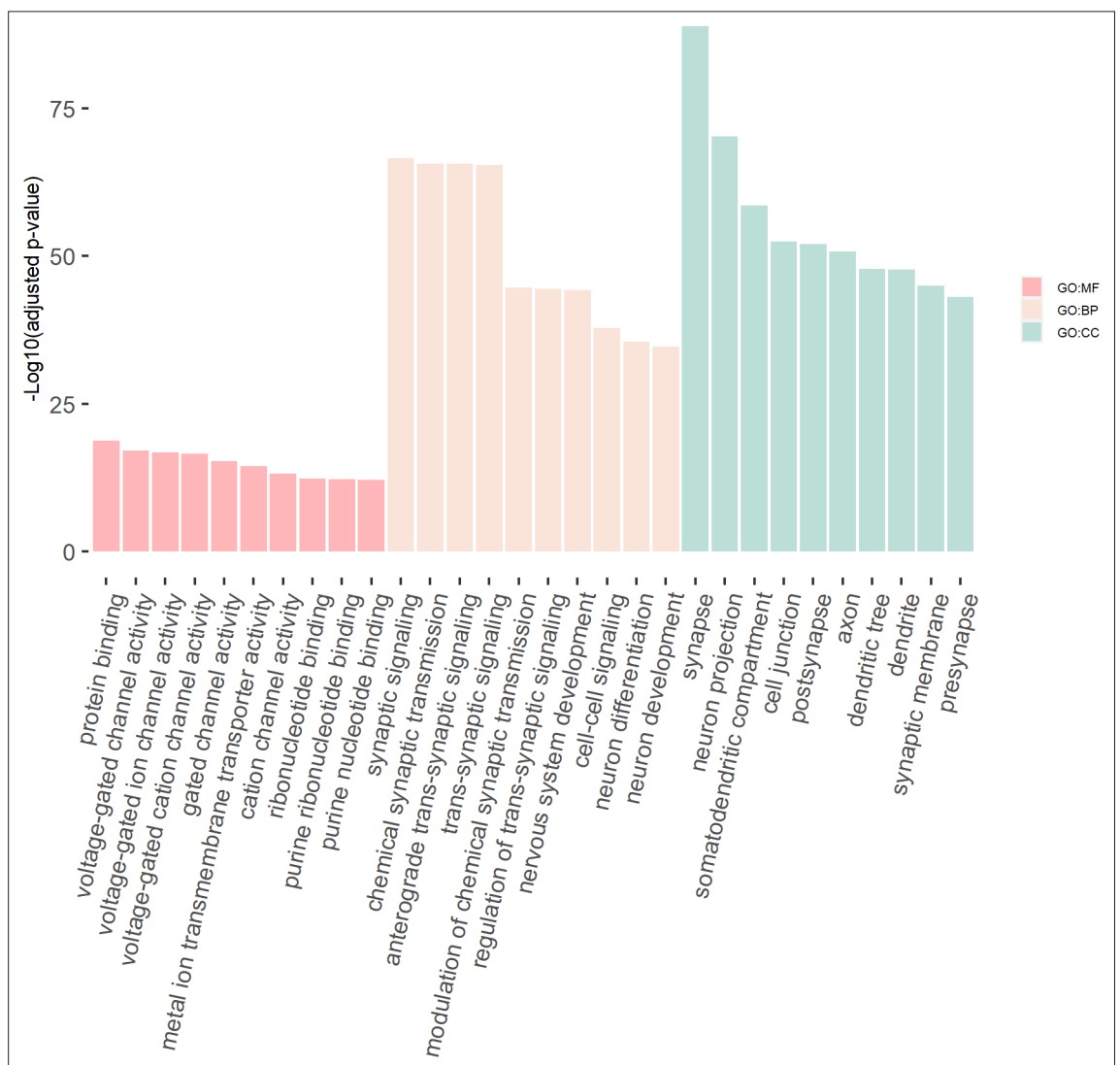

**Figure 8.** The GO enrichment terms of the genes in the consensus module (colored 'turquoise'). The *x*-axis label represents the top (≤10) significant GO enrichment terms (in MF, BP, and CC). The y-axis label represents the negative log10 of the adjusted p-value of each term. See also *Figure 8—source data 1* and *Figure 8—source data 2*.

The online version of this article includes the following figure supplement(s) for figure 8:

**Source data 1.** Genes in the consensus module (colored "turquoise").

**Source data 2.** Enrichment results of the genes in the consensus module (colored 'turquoise').

Brain-Cerebellum were both enriched with a brain consensus module (colored 'turquoise') with statistical significance, that is, adjusted p-value = 0.0046 and 0.0061, respectively. Besides, the potential susceptibility genes identified by eDESE:gene based on the gene-level eQTLs of Brain-Frontal Cortex (BA9) were also enriched (adjusted p-value = 0.024) with the consensus module colored 'turquoise'. However, no enriched consensus module was found for the potential susceptibility genes identified by S-PrediXcan. Moreover, we found that the 'turquoise' consensus module contained 7,726 genes and was enriched with plenty of neuronal and synaptic signaling-related GO terms (*Figure 8*, see details in *Supplementary file 1m*).

**Table 5.** The target genes of the antipsychotics predicted as the potential susceptibility genes by MAGMA, S-PrediXcan, and eDESE.

| Target gene | Models |
| --- | --- |
| DRD2 | eDESE:dist & eDESE:gene & eDESE:isoform & MAGMA |
| ADRA1A | eDESE:isoform |
| CHRM3 | eDESE:gene & eDESE:isoform |
| CHRM4 | eDESE:gene & eDESE:isoform& MAGMA |
| OPRD1 | eDESE:dist |
| GABRD | eDESE:isoform |
| CYP2D6 | eDESE:gene & eDESE:isoform& MAGMA & S-PrediXcan |

## eDESE:isoform can predict the potential susceptibility isoforms of corresponding phenotype-associated tissues

As shown in *Figure 7b*, 55 genes were collectively predicted to be susceptible to schizophrenia by S-PrediXcan, eDESE:gene, and eDESE:isoform. As eDESE:isoform could output susceptibility gene-isoform pairs for corresponding tissues, we further got the corresponding susceptibility isoforms of the 55 genes (see details in *Supplementary file 1n*). Interestingly, we found that the number or type of susceptibility isoforms for a gene varied greatly in the different optimized brain regions. For example, *NAGA* and *CHRNA2* were reported to be associated with schizophrenia by three and seven research papers in the PubMed database, respectively. *ENST00000407991* of *CHRNA2* was significantly associated with schizophrenia only in Brain-Cerebellum, while *ENST00000396398* of *NAGA* was significantly associated with schizophrenia in all the five optimized brain regions. We also found that different isoforms of the same gene were predicted to be significantly associated with schizophrenia in the different optimized brain regions. For example, *SNX19* was reported to be associated with schizophrenia by seven research papers. *ENST00000527116* of *SNX19* in Brain-Anterior cingulate cortex (BA24), *ENST00000528555* of *SNX19* in Brain-Cerebellum, *ENST00000265909* of *SNX19* in Brain-Frontal Cortex (BA9) and Brain-Hippocampus, and *ENST00000526579* of *SNX19* in Brain-Spinal cord (cervical c-1) were significantly associated with schizophrenia, respectively.

The above comparisons suggested that incorporating isoform-level eQTLs can help eDESE predict more potential susceptibility genes than gene-level eQTLs and S-PrediXcan in each optimized brain region. Our results further pointed that eDESE:isoform could help find some novel, biologically sensible susceptibility genes which S-PrediXcan and eDESE:gene cannot find. Moreover, we also found that the potential susceptibility genes identified by eDESE were enriched with a consensus module in the brain weighted gene co-expression network, which was significantly enriched with plenty of biologically sensible GO terms. Based on the isoform-level eQTLs of each phenotype-associated brain region, eDESE:isoform can also help gain insight into the potential susceptibility isoforms. Thus, our results revealed the potential advantages of eDESE:isoform, at least partly, over eDESE:gene and S-PrediXcan.

## The druggability of the potential susceptibility genes

Since drug target genes with genetic support are twice or as likely to be approved than target genes with no known genetic associations (*King et al., 2019*; *Nelson et al., 2015*), we searched the Drug-Bank database (*Wishart et al., 2018*) and found that seven potential susceptibility genes identified by eDESE in total (0.976% for eDESE:dist, 0.632% for eDESE:gene and 0.703% for eDESE:isoform) were the target genes of multiple antipsychotics (*Table 5*). Several popular target genes, such as *DRD2* and *ADRA1A*, were identified by different eDESE models. Besides, we found that three genes (0.485%) of the 619 potential susceptibility genes identified by MAGMA were also the target genes of multiple antipsychotics. One gene (0.256%) of the 391 potential susceptibility genes identified by S-PrediXcan

**Table 6.** The enrichment of drug-gene interaction terms in DGIdb for the susceptibility genes identified by MAGMA, S-PrediXcan and eDESE.

| Models | # of antipsychotics-gene interaction terms | # of total drug-gene interaction terms | Enrichment p* |
|---|---|---|---|
| MAGMA | 57 | 937 | 1.62e-11 |
| S-PrediXcan | 12 | 452 | 0.33 |
| eDESE:dist | 34 | 279 | 1.56e-15 |
| eDESE:gene | 56 | 968 | 1.65e-10 |
| eDESE:isoform | 70 | 1,104 | 8.74e-15 |

*Enrichment p denotes the p-value of the hypergeometric distribution test. See also Table 6—source data 1, Table 6—source data 2, Table 6—source data 3, Table 6—source data 4, and Table 6—source data 5.

The online version of this article includes the following source data for table 6:

**Source data 1.** The drug-gene interaction term results of the potential susceptibility genes of schizophrenia identified by MAGMA in DGIdb.

**Source data 2.** The drug-gene interaction term results of the potential susceptibility genes of schizophrenia identified by S-PrediXcan in DGIdb.

**Source data 3.** The drug-gene interaction term results of the potential susceptibility genes of schizophrenia identified by eDESE:dist in DGIdb.

**Source data 4.** The drug-gene interaction term results of the potential susceptibility genes of schizophrenia identified by eDESE:gene in DGIdb.

**Source data 5.** The drug-gene interaction term results of the potential susceptibility genes of schizophrenia identified by eDESE:isoform in DGIdb.

was the target gene of an atypical antipsychotic (i.e. paliperidone). The results suggested that the three models of eDESE could complement each other to identify more susceptibility genes which could be the target genes of the therapeutic drugs.

To further investigate the drug-gene interactions of the potential susceptibility genes, we searched the Drug Gene Interaction database (DGIdb v4.2.0) (*Freshour et al., 2021*) and kept the drug-gene interaction terms with at least one supported PubMed paper. After the filtration, we kept 30,072 unique drug-gene interaction terms and found 679 unique drug-gene interaction terms for 34 antipsychotics (see details in *Supplementary file 1o*). Then we put the combined potential susceptibility gene list (identified by MAGMA, S-PrediXcan, eDESE:dist, eDESE:gene and eDESE:isoform, respectively) into DGIdb to investigate if the 'antipsychotic'-'potential susceptibility gene' interaction terms were enriched in the known drug-gene interaction database, that is, DGIdb. As shown in *Table 6*, we found that 'anti-psychotic' - 'potential susceptibility gene' interaction terms identified by MAGMA and the three models of eDESE were all significantly enriched in DGIdb. We also found 452 drug-gene interaction terms for the susceptibility gene identified by S-PrediXcan. However, only 12 'antipsychotic'- 'potential suscepti-bility gene' interaction terms were found (hypergeometric distribution test p-value = 0.33).

Moreover, we investigated the potential druggability of the potential susceptibility genes. Among the 42 potentially druggable gene categories in DGIdb, we found that the top potentially druggable category for the potential susceptibility genes identified by MAGMA, S-PrediXcan and eDESE all were the "DRUGGABLE GENOME". The number of "DRUGGABLE GENOME" genes were 168 (27.1%) for MAGMA, 86 (22.0%) for S-PrediXcan, 62 (30.2%) for eDESE:dist, 136 (21.5%) for eDESE:gene and 207 (24.2%) for eDESE:isoform.

Taken together, compared with MAGMA and S-PrediXcan, our results showed that the potential susceptibility genes identified by eDESE were more likely to be the target genes of therapeutic drugs. Besides, the application of eQTLs (especially the isoform-level eQTLs) could aid eDESE in identifying more potentially druggable genes.

## Discussion

In this study, we proposed a multi-strategy conditional gene-based association framework, eDESE, based on a new effective chi-squared statistic (ECS) to identify the potential susceptibility tissues,

genes, and isoforms for complex diseases. Compared with the unconditional association test based on the new ECS and likelihood ratio test, the conditional association test based on the new ECS showed a lower type I error rate and higher statistical power. Except the improved ECS, eDESE has another advantage of using three mapping strategies, that is, mapping based on physical position and the gene-level and isoform-level variant-gene eQTLs associations, to perform the gene-based association analysis. We implemented the three mapping strategies in corresponding three conditional gene-based association models, that is, eDESE:dist, eDESE:gene, and eDESE:isoform.

eDESE:dist and conventional MAGMA both map variants to genes based on their physical distance to perform gene-based association analysis. eDESE:dist differs from MAGMA mainly in that the former is a conditional gene-based association analysis based on the improved ECS. Although eDESE:dist predicted a smaller size of potential susceptibility genes than MAGMA, more than half of the genes identified by eDESE:dist were also identified by MAGMA. The overlapped potential susceptibility genes (accounting for 51.2% in eDESE:dist and 17.0% in MAGMA) were enriched with plenty of neuronal- or synaptic signaling-related GO terms. Similar results of a recent large-scale GWAS were obtained by gene-set analyses (*Trubetskoy et al., 2022*; *Legge et al., 2021*). The comparison of MAGMA and eDESE:dist revealed the advantage of conditional association analysis, that is, removing the redundantly associated genes. In addition, our results showed that the unique potential susceptibility genes identified by eDESE:dist were enriched with more biologically sensible GO terms than MAGMA. Moreover, the proportion of the susceptibility genes identified by eDESE:dist supported by research papers were more than that of MAGMA. Besides, eDESE:dist could produce a potential susceptibility tissue list for complex diseases because of the involvement of gene selective expression analysis.

Nevertheless, eDESE:dist might omit some distal but important variant-gene associations. We thus expanded eDESE by using eDESE:gene and eDESE:isoform. Using a simulation study, we firstly demonstrated that isoform-level eQTLs were more powerful than gene-level eQTLs in association analysis based on the improved ECS. Then taking schizophrenia as an example, we predicted the potential susceptibility tissues by integrating the results of eDESE:dist, eDESE:gene and eDESE:isoform based on the principle of 'wisdom of the crowds' and finally optimized five brain regions. Interestingly, four of the five optimized brain regions (i.e. Brain-Anterior cingulate cortex (BA24), Brain-Cerebellum, Brain-Frontal Cortex (BA9), and Brain-Hippocampus) were also reported by the Schizophrenia Working Group of the Psychiatric Genomics Consortium, in which the genes with high relative specificity for bulk expression were strongly enriched with schizophrenia associations (*Trubetskoy et al., 2022*).

Furthermore, we compared the potential susceptibility genes identified by eDESE:isoform with eDESE:gene and S-PrediXcan, based on the eQTLs of the five optimized brain regions, to demonstrate its potential gains and insights in the biology of schizophrenia. We found that eDESE:isoform could identify more susceptibility genes supported by the published papers and enriched with biologically sensible GO biological process terms. Interestingly, we also found susceptibility genes identified by eDESE:isoform in Brain-Cerebellum and eDESE:gene in both Brain-Cerebellum and Brain-Frontal Cortex (BA9), to be enriched with the same consensus module, which was stunningly enriched with plenty of neuronal- or synaptic signaling-related GO terms. Moreover, to our best knowledge, eDESE:isoform is the first conditional gene-based association model to produce a list of phenotype-associated isoforms (or transcripts) for complex diseases. The comparisons among eDESE:isoform, eDESE:gene, and S-PrediXcan revealed that our model, especially the eDESE:isoform, can gain more potential insights for schizophrenia biology.

To further investigate if eDESE can help gain more insights into the gene druggability, we compared the drug-gene interaction terms and potentially druggable category of the susceptibility genes identified by MAGMA, S-PrediXcan and eDESE. Seven susceptibility genes identified by eDESE were found to be the target genes of multiple FDA-approved antipsychotics, and six (85.7%) of the seven target genes were found by eDESE:isoform. On the other hand, only one and three susceptibility genes identified by S-PrediXcan and MAGMA were found to be the target genes of several FDA-approved antipsychotics. In addition, the 'antipsychotics'-'susceptibility gene' interactions terms for MAGMA and eDESE were both significantly enriched in the known drug-gene interaction database (DGIdb). Moreover, we found that the number of potential susceptibility genes identified by eDESE:isoform and denoted as 'DRUGGABLE GENOME' in DGIdb, were the most among MAGMA, S-PrediXcan, and

eDESE. The comparison revealed that the druggablilty of the potential susceptibility genes identified by eDESE, especially the eDESE:isoform, provided more credible supports for the utility of eDESE.

Our framework might have three potential applications. First, eDESE can be used to predict the potential susceptibility genes and isoforms for other complex diseases. Second, eDESE can help optimize the more biologically sensible phenotype-associated tissues for other complex diseases. Third, the expression profiles used by eDESE can be replaced with other profiles, such as drug-induced gene expression profiles, to investigate the potential drug mechanism of action (MOA).

The present study was also limited by several factors. First, the moderate sample size in GTEx (v8) might lead the gene/isoform-level eQTLs to be underpowered. Future genetic studies based on the increased sample sizes might alleviate this problem. Second, the size of the susceptibility genes identified by the eDESE:gene and eDESE:isoform was a little larger than that of the conventional studies. The main reasons might lie in the gene-level and isoform-level eQTLs selected based on a lenient threshold (p-value < 0.01) to involve more eQTLs in the association analysis, which was taken as the remedy against the underpowered eQTLs. As shown in the present study, conventional MAGMA also identified 619 significant genes. The comparison of MAGMA and eDESE showed the advantages of eDESE in GO enrichment analysis, text-mining analyses and druggability. Although these potential susceptibility genes identified by eDESE lacked systematically experimental validation, we shared these genes in *Supplementary file 1ag and h* and encouraged the follow-up studies to evaluate their functions and roles in the development of schizophrenia. Future studies based on increased sample sizes with a stricter threshold to select the eQTLs can reduce the potential susceptibility gene count, and this is also a topic for our future exploration. Third, the performance of conditional association analysis for fine-mapping can be greatly influenced by the gene orders of entering the analysis. Although the improved effective chi-squared test with eDESE can work well in the real data of schizophrenia, integrating some non-conditional fine-mapping methods, such as SuSiE, with eDESE is worth trying. Fourth, the optimal $c$ value of the effective chi-squared test is still empirical although we derived its range and relevant factors. The optimal $c$ value can be improved to be better suited for other specific application scenarios.

In conclusion, in this study, we proposed a new statistical framework to predict potential susceptibility genes for complex diseases based on the GWAS summary statistics and three different variant-gene mapping strategies. The application of our framework to schizophrenia revealed many novel susceptible and druggable candidate genes. Besides, our results suggested that the usage of isoform-level eQTLs can be an important supplement for the conventional gene-based approaches. The framework was packaged and implemented in our integrative platform KGGSEE. We hope our framework can facilitate researchers to gain more insights into the susceptibility genes, isoforms and tissues for complex diseases.

## Materials and methods
### The new effective chi-squared statistics (ECS) for conditional gene-based association analysis

The effective chi-squared statistics, ECS (*Li et al., 2019*), was improved by using a new correlation matrix of chi-squared statistics, which had two methodological advances to address the potential inflation issue, that is, a type I error-controlled correlation matrix of the observed chi-squared statistics and a non-negative least square solution for the independent chi-squared statistics. Suppose that gene set $A$ contains $n$ loci, to calculate the association p-value of another physically nearby gene (containing $m$ loci) by conditioning on $A$, the first step of the conditional analysis was to produce the effective chi-squared statistics for $A$ ($n$ loci) and all the genes (n + $m$ loci in total). Each locus had a p-value for phenotype association in the GWAS. The p-values can be converted to corresponding chi-squared statistics with the degree of freedom 1. According to *Li et al., 2019*, each locus could be assumed to have a virtually independent chi-squared statistic. An observed marginal chi-squared statistic of a locus was equal to the summation of its virtually independent chi-squared statistic and the weighted virtually independent chi-squared statistic of the nearby loci. The weight was related to the correlation of chi-squared statistics, a key parameter of the analysis. The correlation of chi-squared statistics between two loci was approximated by the absolute value of genotypic correlation to the power of $c$, that is, $|r|^c$. We derived that exponent $c$ ranged from 1 to 2, corresponding to different

noncentrality parameters of a non-central chi-squared distribution (See the derivation in the third section of Materials and methods). The $n$ virtually independent chi-squared statistics of the gene set $A$ could be approximated by a linear transformation of the $n$ observed chi-squared statistics by **Formula (1)**,

$$\begin{bmatrix} \acute{x}_1^2 & d_1 \\ ... & ... \\ \acute{x}_n^2 & d_n \end{bmatrix} \approx \begin{bmatrix} 1 & ... & |r_{1,n}|^c \\ ... & 1 & ... \\ |r_{n,1}|^c & ... & 1 \end{bmatrix}^{-1} \times \begin{bmatrix} x_1^2 & 1 \\ ... & ... \\ x_n^2 & 1 \end{bmatrix} \tag{1}$$

where $\acute{x}_n^2 (\geq 0)$, $d_n (>0)$, $x_n^2$ and $|r_{i,j}|$ denoted a virtually independent chi-squared statistic, degree of freedom of the virtually independent chi-squared statistic, an observed chi-squared statistic and the absolute value of the LD correlation coefficient (approximated by genotypic correlation), respectively. The effective chi-squared statistic $\acute{S}_n$ with the degree of freedom $\acute{d}_n$ of the $n$ loci was then obtained by **Formula (2)**:

$$\begin{cases} \acute{S}_n = \Sigma_{i=1}^n \acute{x}_i^2 \\ \acute{d}_n = \Sigma_{i=1}^n d_i \end{cases} \tag{2}$$

The effective chi-squared statistics ($\acute{S}_{n+m}$) and degree of freedom ($\acute{d}_{n+m}$) of the n + m loci were calculated in the same way. The effective chi-squared statistics of the $m$ loci conditioning on the $n$ loci were approximated by **Formula (3)**,

$$\acute{S}_{m|n} = \acute{S}_{n+m} - \acute{S}_n \tag{3}$$

with the degree of freedom $\acute{d}_{m|n} = \acute{d}_{n+m} - \acute{d}_n$. Because $\acute{d}_{m|n}$ was no longer an integer, we used the Gamma distribution to calculate the p-values. Given the above statistics and degree of freedom, the p-value was equal to $F(x \geq \frac{\acute{S}_{m|n}}{2}; \frac{\acute{d}_{m|n}}{2}, 2)$, where the $F(x)$ function was the cumulative distribution function of a Gamma distribution.

Because the virtually independent chi-squared statistics and degree of freedom were expected to be larger than 0, we adopted a sequential coordinate-wise algorithm to approximate them (*Franc et al., 2005*). This algorithm avoided unstable solutions in **Formula (1)** due to stochastic errors in the correlation matrix and observed chi-squared statistics.

## Choose the favorable $c$ value for the correlation matrix of chi-squared statistics

After multiple approximations, the analytic solution for the exponent $c$ in **Formula (1)** was still difficult to obtain. We proposed a grid search algorithm to find a favorable value of exponent $c$ to control the type I error rates of ECS. The type I error rate was examined by the departure between the obtained and the theoretical (under the uniform distribution) top 1% p-values given a $c$ value, measured as mean log fold change (MLFC) (*Tokheim et al., 2016*). In the grid search process, we increased $c$ from 1.00 to 2.00 by an interval of 0.01. The $c$ value leading to the minimal MLFC was defined as the favorable $c$ value. We considered in total 84 parameter settings, i.e., a combination of three different sample sizes (10,000, 20,000 and 40,000) and 14 different variant number (10, 30, 50, 80, 100, 125, 150, 200, 250, 300, 400, 500, 800, and 1000) for both binary and continuous traits, respectively. For a parameter setting, 40,000 datasets were simulated and used to produce p-values to determine the favorable $c$ value. A region on chromosome 2 [chr2: 169,428,016–189,671,923] was randomly drawn for the simulation. The allele frequencies and LD structure of variants in the European panel of the 1000 Genomes Project were used as a reference to simulate genotype data by the HapSim algorithm (*Montana, 2005*). Each subject was randomly assigned a phenotype value under the null hypothesis according to the Bernoulli or Gaussian distribution. The Wald test, which encoded the major and minor allele as 0 and 1 under either logistic regression or linear regression, was used to produce the association p-value at each variant. The p-values of the variants were then analyzed by the effective chi-squared test for the gene-based association analysis.

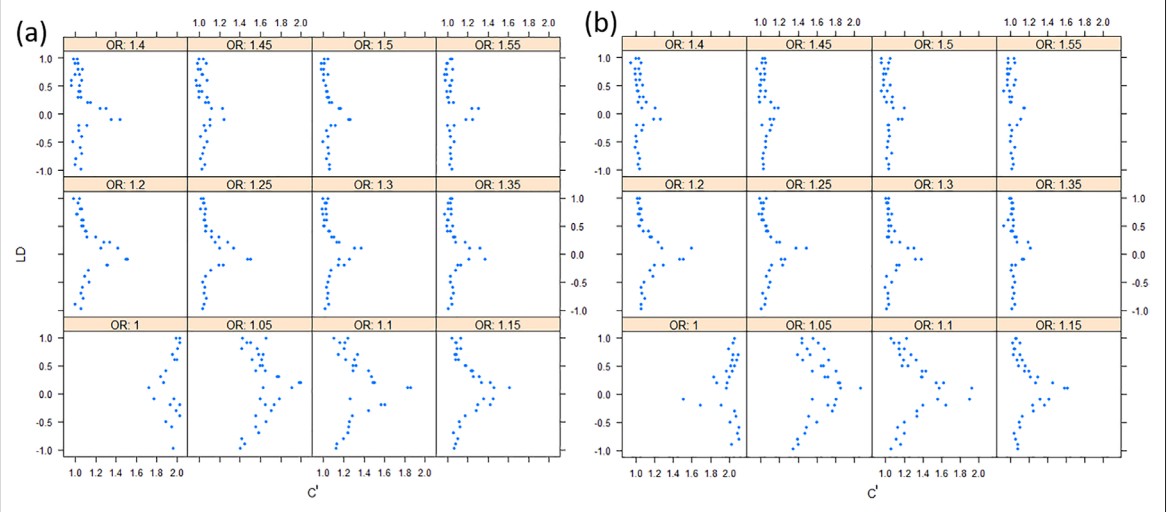

**Figure 9.** The effect sizes of variants to the correlation of chi-squares. (**a**) Under the additive model; (**b**) Under the multiplicative model. The allele frequencies are assigned randomly.

## Approximate the correlation of chi-square statistics under the alternative Hypothesis

Let two normal random variables $X \sim N(\mu_1, \sigma_1^2)$ and $Y \sim N(\mu_2, \sigma_2^2)$ have covariance **a**. Note that a squared normal random variable has non-central chi-square distribution, and the squared mean of the variable is called noncentrality parameter. The two variables can also be factorized as $X = \mu_1 + \sigma_1 U$, $Y = \mu_2 + \sigma_2(\rho U + \sqrt{1 - \rho^2} V)$ where $U, V \overset{iid}{\sim} N(0,1)$ and $\rho = a/(\sigma_1 \sigma_2)$ .

Then we can calculate the correlation of the two non-central chi-square variables $X^2$ and $Y^2$ by the factorized variables, $cor(X^2, Y^2) = \frac{2\mu_1\mu_2 a + a^2}{\sqrt{2\mu_1^2\sigma_1^2 + \sigma_1^4}\sqrt{2\mu_2^2\sigma_2^2 + \sigma_2^4}}$ . Suppose **X** is the z-score of the true casual variant, and **Y** is the z-score of a non-functional variant in LD (coefficient **r**) with the causal variant. One can assume $\mu_2 = r\mu_1$ , $\sigma_1^2 = 1$ , $\sigma_2^2 = 1$ and **a** = **r**. Therefore, the correlation of $X^2$ and $Y^2$ can be simplified as, $c' = \frac{2r^2\mu_1^2 + r^2}{\sqrt{2\mu_1^2 + 1}\sqrt{2\mu_1^2 r^2 + 1}}$ .

Under the null hypothesis, $\mu_1 = 0$ then $cor(X^2, Y^2) = r^2$ . Under the alternative hypothesis, $\mu_1^2$ is the noncentrality parameter ($\lambda$). According to Sham and Purcell (***Sham and Purcell, 2014***), the noncentrality parameter can be approximated by the following Formula for a quantitative phenotype, $\mu_1^2 = \lambda = N\frac{2\beta^2 p(1-p)}{Residual\ variance\ of\ phenotype}$, where N, p and $\beta$ are the sample sizes, allele frequency and the regression coefficient (corresponding to effect size), respectively. This Formula can be extended for qualitative phenotype under a liability threshold model. N is very large for a large sample, and the noncentrality parameter will be large for common variants. Therefore, the correlation of $X^2$ and $Y^2$ converges to **|r|** as $\lambda \to \infty$, $cor(X^2, Y^2) = \frac{2|r|}{\sqrt{2 + \frac{1}{\mu_1^2}}\sqrt{2 + \frac{1}{r^2\mu_1^2}}} + \frac{r^2}{\sqrt{2\mu_1^2 + 1}\sqrt{2r^2\mu_1^2 + 1}} = \frac{2|r|}{\sqrt{2 + \frac{1}{\lambda}}\sqrt{2 + \frac{1}{\lambda r^2}}} + \frac{r^2}{\sqrt{2\lambda + 1}\sqrt{2\lambda r^2 + 1}} \to |r|$.

Overall, the correlation between the two (non-central) chi-square ranges can be approximated as

$|r|^{c'}$, in which the $c'$ ranges from 1 to 2. In simulation studies, we also showed that as the relative risk

ratio increased to 1.5 in a moderate sample of 1000 cases and 1000 controls, the correlation of $X^2$ and $Y^2$ became close to **|r|** (See ***Figure 9***).

## The conditional gene-based association analysis for genome-wide association study

In a GWAS, the p-values of all genes in the unconditional gene-based association analysis were firstly calculated using the above effective chi-squared statistics. Then, for a given p-value cutoff, the significant genes were extracted and subjected to the conditional gene-based association analysis. Multiple significant genes in an LD block were conditioned one by one in a pre-defined order. The order of the

gene was defined according to the unconditional *p*-value of the gene. Here the genes within 5 Mb were assigned into the same LD block. The conditional *p*-value of the first gene was defined as its unconditional *p*-value. The conditional *p*-value of the second gene was obtained by conditioning on the first gene, and that of the third gene was obtained by conditioning on the first two genes. The conditional *p*-values of subsequent genes were calculated according to the same procedure.

## Investigate the type I error and power of the conditional gene-based association analysis

Independent computer simulations based on a different reference population (i.e., EAS) and genomic regions were performed to investigate type I error and the power of the conditional gene-based association analysis. To approach the association redundancy pattern in realistic scenarios, we used the real genotypes and simulated phenotypes. The high-quality genotypes of 2,507 Chinese subjects from a GWAS were used (**Kung et al., 2010**), and phenotypes of subjects were simulated according to the genotypes under an additive model. Given total variance explained by *n* independent variants, $V_g$, the effect of an allele at a bi-allelic variant was calculated by $a = \sqrt{V_g / \left[ \sum_{i=1}^{n} 2P_{A_i} \left(1 - P_{A_i}\right) \right]}$ , where $P_{A_i}$ was the frequency of alternative alleles. The total expected effect *A* of a subject was equal to a*[the number of alternative alleles of all the n variants]. Each subject's phenotype was simulated by *P = A + e*, where *e* was sampled from a normal distribution *N*(0, 1-$V_g$). We randomly sampled three pairs of genes, i.e., *SIPA1L2* vs. *LOC729336*, *CACHD1* vs. *RAVER2*, and *LOC647132* vs. *FAM5C*, representing three scenarios where the nearby gene (i.e., the first gene) had similar (*SIPA1L2* vs. *LOC729336*), larger (*CACHD1* vs. *RAVER2*) and smaller (*LOC647132* vs. *FAM5C*) variant number than the target gene (i.e., the second gene) in terms of SNP number, respectively. The target gene had no QTLs in the type I error investigation, while the nearby gene had one or two QTLs. In the investigation of the statistical power, both the target and nearby genes had QTLs.

The likelihood ratio test based on the linear regression was adopted for power comparison to perform the conditional gene-based association analysis with raw genotypes. In the full model, genotypes of all SNPs encoded as 0, 1, or 2 according to the number of alternative variants entered the regression model as explanatory variables. In the subset model, the SNPs of the nearby genes entered the regression model. The calculation of the likelihood ratio test was performed according to the conventional procedure. The R packaged "lmtest" (version 0.9.37) was adopted to perform the likelihood ratio test.

## Investigate the power and type I error of gene-level eQTLs and isoform-level eQTLs in gene-based association analysis

The same region on chromosome 2 [chr2: 169,428,016–189,671,923] was considered for the simulation. In the EUR panel of 1000 Genomes Project (**Auton et al., 2015**), this region contains 1,600 common variants (MAF >0.05). Genotypes of the variants were simulated given allelic frequencies and LD correlation matrix according to the HapSim algorithm (**Montana, 2005**). Phenotypes were simulated under a polygenic model of random effect (**Bulik-Sullivan et al., 2015**). According to severe LD pruning ($r^2$ <0.01), eighty-two independent variants were extracted from the 1,600 variants. The SNPs' genotypes (*s*) contributing to the phenotypes were then standardized as, $g = \left(s - 2q\right) / \sqrt{2q \left(1 - q\right)}$ , where *q* was the allele frequency of the alterative allele. Phenotypes were simulated under a polygenic model of random effect (**Bulik-Sullivan et al., 2015**). We assumed that 40% of the independent causal variants ($m_X$) regulated gene expression (total heritability $h_X^2$). The expression of a gene (X) was simulated according to **Formula (4)**:

$$X = \sum_{i=1}^{m_X} g_i \beta_{X,i} + \epsilon_X \tag{4}$$

Where $\beta_{X,i} \sim N(0, h_X^2/m_X)$ and $\epsilon_X \sim N(0, 1 - h_X^2)$.

The gene expression then contributed $\delta$ to a phenotype (Y). The phenotype value was simulated according to the **Formula (5)**:

$$Y = \delta X + \sum_{i=1}^{m_Y} g_i \beta_{Y,i} + \epsilon_Y \tag{5}$$

where Y was a continuous phenotype, $\beta_{Y,i} \sim N(0, h_Y^2/m_Y)$ and $\epsilon_Y \sim N(0, 1 - h_Y^2 - \delta^2)$ . For a binary phenotype, a cutoff $t$ was set according to a given disease prevalence $K$ under a standard normal distribution and the liability threshold model (*Gillett et al., 2018*). Subjects with simulated Y values ≥ $t$ were set as patients, and others were set as normal controls.

When a gene had multiple isoforms, we assumed that one of the isoforms was associated with phenotype, and we simulated the expression values of the isoform according to **Formula (5)**. The expression values of the remaining isoforms were simulated by the standard normal distribution $N(0,1)$. The expression profile of a gene with multiple isoforms was averaged by all the isoforms belonging to the gene. The gene-level eQTLs and isoform-level eQTLs were examined by the Wald test under the linear regression model. The variant-phenotype association analysis was performed based on the conventional association analysis procedure, and the statistical significance cutoff was p-value < 0.001. For each parameter setting, $t$ (an integer) datasets were simulated, and the power and type I error were estimated by $m/t$, in which $m$ was the number of datasets with significant $p$-values for testing δ.

## Genome-wide association study of schizophrenia

The schizophrenia GWAS included 53,386 cases and 77,258 controls of European ancestry samples (hg19 assembly). Genotypes in the CEU panel from the 1000 Genomes Project were used to correct for the relatedness of the summary statistics. The variants in the major histocompatibility complex (MHC) region, i.e., chr6:27,477,797–34,448,354, were excluded in the present study because of the high polymorphism. Detailed descriptions of population cohorts, quality control methods and association analysis methods can be found in reference (*Trubetskoy et al., 2022*).

## The Genotype-Tissue Expression (GTEx) Project

The GTEx project (release v8, RRID:SCR_013042) created a resource including whole-genome sequence data and RNA sequencing data from ~900 deceased adult donors (*Consortium, 2020*). Four tissues or cell types (i.e., whole blood, stomach, pancreas and pituitary) were filtered out in the following analyses because of their small sample sizes or weak correlations with most other tissues.

## g:Profiler and Hetionet

All GO enrichment analyses were performed by g:Profiler (*Raudvere et al., 2019*). g:Profiler (RRID:SCR_006809) is based on Fisher's one-tailed test. The statistical *p*-value is multiple testing-corrected. The GO enrichment analysis uses the set of all annotated protein-coding genes for *Homo sapiens* (Human) as background. Significant GO terms were filtered by the threshold of "Padj" < 0.05. The bar plots of GO enrichment terms were drawn based on R-4.0.3.

Hetionet (*Himmelstein et al., 2017*) integrates relationships among genes, compounds, diseases, and more from 29 different databases. It can help researchers refine their phenotype-gene associations by considering anatomies, biological processes, side-effects, symptoms and more.

## Construct the weighted gene co-expression network for brain and perform the consensus analysis

The thirteen brain regions' fully processed, filtered, and normalized gene-level expression profiles in GTEx (v8) were used. Consensus network analysis and module identification were performed based on the "WGCNA" (v1.69, RRID:SCR_003302) (*Langfelder and Horvath, 2008*). For each dataset, WGCNA was performed to build a signed gene co-expression network following the standard procedure, and the soft-threshold was finally set as 12 after testing a series of soft-threshold powers (ranging from 2 to 20). As for constructing the block-wise consensus modules, the hierarchical cluster tree in the co-expression network was cut into gene modules using the dynamic tree cut algorithm with a minimum module size of 30 genes (*Langfelder et al., 2008*). The parameter of "networkCalibration" was set as "single quantile". The "consensusQuantile" and "calibrationQuantile" were both set as 0.95. The parameter of "deepSplit" was set as 3. Other parameters were used as recommended by WGCNA. The co-expression analysis and consensus clustering analysis produced eighteen consensus modules, in which the module sizes ranged from 41 to 7,726. The significant consensus modules were filtered by the threshold of "Padj" < 0.05.

## Drug Gene Interaction Database (DGIdb)

DGIdb (v4.2.0, RRID:SCR_006608) provides a resource of genes that have the potential to be druggable and contains two main classes of druggable genome information (*Freshour et al., 2021*). The first class includes genes with known drug interactions, and the other includes genes that are potentially druggable according to their membership in gene categories associated with druggability.

## PubMed text-mining analysis

To find supports from the published research, we performed a text-mining analysis based on PubMed database on August 27th, 2021 using a java script. We put each gene symbol name or its synonyms into the PubMed database with the items of "((schizophrenia[tiab]+ OR + Schizophrenia[tiab]+ OR + SCZ[tiab])+ AND + (genename[tiab])+ AND + (gene[tiab]+ OR + genes[tiab]+ OR + mRNA[tiab]+ OR + protein[tiab]+ OR + proteins[tiab]+ OR + transcription[tiab]+ OR + transcript[tiab]+ OR + transcripts[tiab]+ OR + expressed[tiab]+ OR + expression[tiab]+ OR + expressions[tiab]+ OR + locus[tiab]+ OR + loci[tiab]+ OR + SNP[tiab]))&datetype = edat&retmax = 100". The java script output a file with the first column representing gene name, the second column representing the synonym of the gene name and the last column representing the PubMed ids of hit papers.

## Identify the potentially phenotype-associated tissues of schizophrenia

To estimate the potentially phenotype-associated tissues, eDESE:dist, eDESE:gene and eDESE:isoform were used, respectively. The Genotypes in the EUR panel from the 1000 Genomes Project (phase 3) were downloaded from IGSR and used as reference genotype data. Three columns, that is., chromosome identifier, base-pair position and p-value in the GWAS summary statistics, were used. SNPs with minor allele frequency (MAF) less than 0.05 were excluded. Genes approved by HGNC (*HGNC Database, H.G.N.C.H, 2021*) were included in the following analyses. The multiple testing adjustment method was Bonferroni correction, and the cutoff for the adjusted p-value was set as p-value < 0.05. The threshold for filtering the significant phenotype-associated tissues was set as $\alpha$ = 0.05/50 = 1e-3. The detailed commands of eDESE to identify the potential phenotype-associated tissues are described on the KGGSEE website and the original paper (*Jiang et al., 2019*).

## Identify gene-level and isoform-level eQTLs

The present study focused on the *cis*-eQTLs. Specifically, two files (expression profiles and corresponding genotype data file from the European ancestry subjects in GTEx v8) were put into KGGSEE to produce the gene/isoform-level eQTLs for each tissue. Two levels (gene-level and isoform-level) expression profiles of 50 tissues were downloaded from the GTEx v8 project and were normalized as GTEx did (https://gtexportal.org/home/documentationPage). Genes/isoforms were selected based on TPM >0.1 and read count ≥6 in at least 20% of all samples. Only variants with MAF ≥0.05 were included in the eQTLs identification. GTEx v8 is based on the human reference genome GRCh38/hg38. Thus, to be consistent with the GWAS results of schizophrenia (hg19 assembly), we converted the GRCh38/hg38 coordinates into hg19 by using the UCSC LiftOver (*Hinrichs et al., 2006*). Variants with Hardy-Weinberg disequilibrium (HWD) test p-value < 1.0e-3 were filtered out. The mapping window was defined as 1 Mb up- and downstream of the gene boundary. The covariates used in eQTLs identification include donor sex, age and death classification. The threshold for selecting the gene-level/isoform-level eQTLs was p-value < 0.01.

## Estimate the potential susceptibility genes and isoforms

For eDESE:dist, if a variant was within ±5 kb around a gene boundary, the variant will be mapped to the gene according to a gene model, for example, RefSeqGene. For eDESE:gene and eDESE:isoform, a variant was mapped to a gene or isoform if the variant is a gene/isoform-level eQTL of the gene or isoform.

eDESE adopted the iterative procedure from DESE (*Jiang et al., 2019*). In the first iterative step, genes with smaller p-values generated from the unconditional association analysis (based on the improved ECS) were given higher priority to enter the following conditional gene-based association analysis. We then dealt with the three models of eDESE in different ways. For eDESE:dist and eDESE:gene, the order of a gene entering the conditional gene-based association analysis was determined by its p-value generated from the unconditional association analysis. For eDESE:isoform, assume that

gene $A$ has $m$ isoforms. Each isoform could get a p-value generated from the unconditional association analysis, representing the overall statistical significance of all isoform-level eQTLs (simultaneously variants) associated with this isoform. If the isoform with the smallest p-value was isoform $a$ with p-value $p_a$ among the $m$ isoforms, we only kept the most significant isoform, that is, isoform $a$ of gene $A$ for the following analyses. The p-value for isoform $a$ was adjusted to $m* p_a$ before entering the following conditional gene-based association analysis.

The second step was to compute the robust-regression z-score of each gene/isoform (see details in reference *Jiang et al., 2019*). The Wilcoxon rank-sum test was then performed by using the robust-regression z-score of the associated and not-associated gene/isoform set (generated by the first step) in each tissue.

In the third step, all genes/isoforms were ranked in descending order based on their tissue-selective expression score, which was computed based on the rank of this gene's or isoform's robust-regression z-score and the p-value of the Wilcoxon rank-sum test (generated by the second step).

In the following iteration, genes/isoforms with higher tissue-selective expression scores (generated in the third step) were given higher priority to enter the conditional gene-based association analysis (back to the first step). The above three iterative steps would not stop until the p-values of the Wilcoxon rank-sum test did not change almost. Then corresponding significant genes/isoforms and tissues were deemed to be potentially associated with the phenotype. More details about the iterative procedure can be found in the original papers (*Jiang et al., 2019*).

The detailed commands, input and output datasets of eDESE can be seen on the KGGSEE website. The bar plot of the comparison of potential susceptibility genes was drawn based on R-4.0.3. The venn diagram was drawn based on a web app Venny 2.1.0.

## MAGMA

MAGMA (RRID:SCR_005757) is a popular tool for gene and generalized gene-set analysis based on the GWAS summary statistics. Here the parameters and options were used as recommended by MAGMA (v 1.09). Annotation analysis was performed based on the SNP and gene location files (hg19, build 37). The SNP location information was extracted from the GWAS summary statistics file of schizophrenia. Both gene location and reference data were downloaded from the MAGMA website. An SNP was mapped to a gene if the SNP was in the window of ±5 kb around the gene boundary (same as eDESE:dist). Then the gene analysis was performed based on the annotation results and reference data file which was created from Phase 3 of 1000 Genomes of the European population in reference to the human genome (build 37). Multiple testing was corrected by using Bonferroni correction. Significant genes were filtered by "Padj" < 0.05.

## S-PrediXcan

S-PrediXcan is command-line based and implemented with python environment and mainly uses summary statistics. To estimate the disease-associated genes of each tissue, we prepared three input files, that is, schizophrenia GWAS summary statistics file, a transcriptome prediction model database file and a file with the covariance matrices of the SNPs within each gene model (*Barbeira et al., 2018*; *Gamazon et al., 2015*; *Barbeira et al., 2021*). Here, GTEx-based tissues and 1000 Genomes covariances precalculated data from the PredictDB repository were downloaded (http://predictdb.org), and the MASHR-based model based on the expression data of GTEx v8 release was used. Other options and parameters were used as recommended. Multiple testing was corrected by using Bonferroni correction. Significant genes in each tissue were filtered by "Padj" < 0.05.

## Data availability statement

All the data used in this study are from public resources. The source data files for the main figures and tables in the manuscript have been provided and are specified in Source Data. The annotations of drug-gene interaction terms are publicly available in Drug Gene Interaction (DGIdb v4.2.0) database in https://dgidb.org. The information on FDA-approved antipsychotics was extracted from DrugBank 5.1.1, which can be freely downloaded from https://go.drugbank.com/releases/5-1-1/downloads/all-full-database with a simple registration for academic users. The functional enrichment analyses were performed by g:Profiler in https://biit.cs.ut.ee/gprofiler. Hetionet v1.0 can be freely accessed at https://het.io/. Venny is in https://bioinfogp.cnb.csic.es/tools/venny/index.html. MAGMA and corresponding

reference data were freely downloaded from https://ctg.cncr.nl/software/magma. S-PrediXcan was freely downloaded from https://github.com/hakyimlab/MetaXcan (*hakyimlab, 2021*) copy archived at swh:1:rev:cfc9e369bbf5630e0c9488993cd877f231c5d02e. The source code of eDESE (including eDESE:dist, eDESE:gene and eDESE:isoform) is implemented in KGGSEE and can be publicly available in http://pmglab.top/kggsee/#/.The custom scripts used in this study can be freely accessed at https://github.com/pmglab/eDESE (*pmglab, 2021*) copy archived at swh:1:rev:68fbbe429f23011f-544cdd34ce09c98a2540f68b (*Li and Li, 2021*).

## Acknowledgements

We thank the GTEx Consortium and 1000 Genomes Projects for providing access to the data and thank the GTEx Consortium for the code to preprocess and normalize the expression profiles. We also appreciate the authors of the schizophrenia study and the Schizophrenia Working Group of the Psychiatric Genomics Consortium for sharing their GWAS summary statistics. Finally, we also thank the authors of E-MAGMA for providing their preprocessed tissue-specific eQTLs of GTEx (v8). This work was funded by the National Natural Science Foundation of China (31771401, 32170637 and 31970650), National Key R&D Program of China (2018YFC0910500 and 2016YFC0904300), Science and Technology Program of Guangzhou (201803010116), Guangdong project (2017GC010644), Department of Science and Technology of Guangdong Province (2018B030322006).

## Additional information

### Funding

| Funder | Grant reference number | Author |
|---|---|---|
| National Natural Science Foundation of China | 31771401 | Miaoxin Li |
| National Key Research and Development Program of China | 2018YFC0910500 and 2016YFC0904300 | Miaoxin Li |
| Science and Technology Program of Guangzhou | 201803010116 | Miaoxin Li |
| Guangdong project | 2017GC010644 | Miaoxin Li |
| National Natural Science Foundation of China | 31970650 | Miaoxin Li |
| National Natural Science Foundation of China | 32170637 | Miaoxin Li |
| Department of Science and Technology of Guangdong Province | 2018B030322006 | Miaoxin Li |

The funders had no role in study design, data collection and interpretation, or the decision to submit the work for publication.

### Author contributions

Xiangyi Li, Data curation, Formal analysis, Investigation, Methodology, Validation, Writing – original draft, Writing – review and editing; Lin Jiang, Data curation, Formal analysis, Investigation, Methodology, Visualization, Writing – original draft, Writing – review and editing; Chao Xue, Data curation, Methodology, Writing – review and editing; Mulin Jun Li, Data curation, Formal analysis, The gene/isoform-level eQTLs were calculated by Mulin Jun Li who has the permission to access the GTEx genotype data; Miaoxin Li, Conceptualization, Data curation, Methodology, Software, Supervision, Writing – review and editing

### Author ORCIDs

Xiangyi Li http://orcid.org/0000-0002-9408-3016
Miaoxin Li http://orcid.org/0000-0002-4733-0109

**Decision letter and Author response**
Decision letter https://doi.org/10.7554/eLife.70779.sa1
Author response https://doi.org/10.7554/eLife.70779.sa2

## Additional files

### Supplementary files

• Supplementary file 1. For eDESE analyses. (a) The schizophrenia potential susceptibility genes identified by eDESE:dist. (b) The GO enrichment results based on the overlapped susceptibility genes identified by MAGMA and eDESE:dist. (c) The PubMed search hits of the schizophrenia potential susceptibility genes identified by MAGMA and eDESE:dist. (d) Tissue significances generated by eDESE:dist based on the gene-level and isoform-level expression profiles. (e) Tissue significances generated by eDESE:gene/isoform using the gene-level and isoform-level eQTLs of Muscle Skeletal and Skin Sun Exposed Lower Leg. (f-h) The schizophrenia potential susceptibility genes identified by S-PrediXcan, eDESE:gene and eDESE:isoform, respectively. (i-k) The PubMed search hits of the combined susceptibility gene set of schizophrenia identified by S-PrediXcan, eDESE:gene and eDESE:isoform, respectively. (l) The GO enrichment results based on the susceptibility genes exclusively predicted by eDESE:isoform. (m) The genes in the consensus module (colored "turquoise") of the brain weighted gene co-expression network. (n) The potential susceptibility isoforms of the 55 overlapped genes. (o) The FDA-approved antipsychotics included in the DGIdb (v4.2.0).

• Transparent reporting form

### Data availability

All the data used in this study are from public resources. The source data files for the main figures and tables in the manuscript have been provided and are specified in source data. The annotations of drug-gene interaction terms are publicly available in Drug Gene Interaction database (DGIdb v4.2.0) in https://dgidb.org/. The information on FDA-approved antipsychotics was extracted from DrugBank 5.1.1, which can be freely downloaded from https://go.drugbank.com/releases/5-1-1/downloads/all-full-database with a simple registration for academic users. The functional enrichment analyses were performed by g:Profiler in https://biit.cs.ut.ee/gprofiler. Hetionet v1.0 can be freely accessed at https://het.io/. Venny is in https://bioinfogp.cnb.csic.es/tools/venny/index.html. MAGMA and corresponding reference data were freely downloaded from https://ctg.cncr.nl/software/magma. S-PrediXcan was freely downloaded from https://github.com/hakyimlab/MetaXcan (copy archived at swh:1:rev:cf-c9e369bbf5630e0c9488993cd877f231c5d02e). The source code of eDESE (including eDESE:dist, eDESE:gene and eDESE:isoform) is implemented in KGGSEE and can be publicly available in http://pmglab.top/kggsee/#/.The custom scripts used in this study can be freely accessed at https://github.com/pmglab/eDESE (copy archived at swh:1:rev:68fbbe429f23011f544cdd34ce09c98a2540f68b).

The following datasets were generated:

| Author(s) | Year | Dataset title | Dataset URL | Database and Identifier |
|---|---|---|---|---|
| Li X, Jiang L, Xue C, Li M, Mj Li | 2021 | Gene-level eQTLs (hg19) based on GTEx v8 | https://figshare.com/articles/dataset/EUR_gene_eqtl_hg19_tar_gz/16959604 | figshare, 10.6084/m9.figshare.16959604 |
| Li X, Jiang L, Xue C, Li M, Mj Li | 2021 | Isoform-level eQTLs (hg19) based on GTEx v8 | https://figshare.com/articles/dataset/EUR_transcript_eqtl_hg19_tar_gz/16959616 | figshare, 10.6084/m9.figshare.16959616 |

The following previously published datasets were used:

| Author(s) | Year | Dataset title | Dataset URL | Database and Identifier |
|---|---|---|---|---|
| The Schizophrenia Working Group of the Psychiatric Genomics Consortium, Ripke S, Walters JTR, O'Donovan MC | 2022 | GWAS summary statistics of schizophrenia | https://www.med.unc.edu/pgc/download-results/ | PGC, SCZ2022 |
| GTEx Consortium | 2020 | RNA-Seq Data | https://gtexportal.org/home/datasets#filesetFilesDiv13 | GTEx v8, RNA-Seq-Data |
| Barbeira AN, Bonazzola R, Gamazon ER, Liang Y | 2021 | MASHR-based models for eQTLs | https://zenodo.org/record/3518299/files/mashr_eqtl.tar?download=1 | PredictDB, mashr_eqtl.tar |
| Gerring ZF, Mina-Vargas A, Martin NG, Gamazon ER, Derks M | 2021 | emagma_annot_1.tar.gz | https://github.com/eskederks/eMAGMA-tutorial/blob/gtex_v8/emagma_annot_1.tar.gz | Github, emagma_annot_1.tar.gz |
| 1000 Genomes Project Consortium | 2015 | Genotype data from the 1000 Genomes Project (phase 3) | http://ftp.1000genomes.ebi.ac.uk/vol1/ftp/ | IGSR, 1000genomes |

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
