## [Editor Report]

This manuscript describes a new method of identifying disease-relevant risk genes from genome wide association studies by using a conditional gene-based method (named eDESE). The authors apply this new approach in an analysis of schizophrenia datasets, and identify meaningful biological processes and potential drug repurposing candidates. Thus, this new method could provide improved gene prioritization for fine-mapping and functional studies of specific diseases.

---

## [Decision Letter]

**Decision letter after peer review:**

Thank you for submitting your article "MCGA: a multi-strategy conditional gene-based association framework integrating with isoform-level expression profiles reveals new susceptible and druggable candidate genes of schizophrenia" for consideration by *eLife*. Your article has been reviewed by 3 peer reviewers, and the evaluation has been overseen by a Reviewing Editor and Molly Przeworski as the Senior Editor. The reviewers have opted to remain anonymous.

Essential revisions:

1) Systematic comparisons with existing methods are needed to demonstrate the potential gain in biological insight from this method. How do the results from MGCA_Dist, MGCA_eQTL, and MGCA_isoQTL compare with MAGMA? Can you include a direct comparison of MGCA with MAGMA, including the gene-based results, biological pathway enrichments, and drug repositioning analyses? Can the authors compare MGCA_eQTL to other eQTL-based approaches such as S-PrediXcan?

2) When comparing MCGA-eQTL and MCGA-sQTL, only power is considered. The authors should include analyses to demonstrate the performance in control for false positives.

3) When choosing a favorable exponent value c (1.432 chosen in the study), the authors found that the c value is robust to trait type, sample size or variant size, but the authors didn't explain what factors affect the choosing of c. Considering the potential application of MCGA method in other studies, the authors should explain what factor affects c value, and provide the guidance how to choose an optimal c. Can the authors show via simulations the relationship between the coefficient used to adjust the chi square statistic correlations, $c$, and the strength of association signals?

*Reviewer #1 (Recommendations for the authors):*

– How do the results from MGCA_Dist, MGCA_eQTL, and MGCA_isoQTL compare with MAGMA? Can you include a direct comparison of MGCA with MAGMA, including the gene-based results, biological pathway enrichments, and drug repositioning analyses?

– Similarly, can you compare MGCA_eQTL to other eQTL-based approaches such as S-PrediXcan? I would like to see a systematic comparison with existing methods to see if novel biological insights can be derived from your results.

– Does MCGA provide information on tissue-specific eQTLs/isoQTLs? This information is available in GTEx but is not reported for association results in supplementary tables. Tissue-specific effects for each tissue might improve the interpretation of results.

– Can you provide more information on WGCNA results? Are MCGA susceptibility genes enriched in a particular co-expression module? Are synaptic signalling related biological pathways enriched in a particular module? How did the network modules compare across brain tissues? Is it possible to build a consensus network for the top five brain regions (or indeed all brain regions in GTEx) to simplify the results?

– The drug repositioning analysis could be improved by drawing on comprehensive drug-gene information from hetionet (https://doi.org/10.7554/*eLife*.26726.001). This resource might further refine some of your top associations by accounting for anatomies, biological processes, side-effects, and symptoms.

*Reviewer #2 (Recommendations for the authors):*

GTEx v8 data contains samples from multiple tissues, and the significant gene-trait association in some tissue may be non-causal driven by gene expression correlation with causal tissues. It should be interesting to come up a new method to remove those false positive.

*Reviewer #3 (Recommendations for the authors):*

My major concern is a lack of innovation as a new method. MCGA is based on DES, which integrates expression data with GWAS; and DESE relies on ECS to compute gene-level association evidence. Both DESE and ECS were developed by the authors in the past and published elsewhere. The only innovation in the MCGA methodology seems the improved control of type I error by formally assessing the null distribution of the conditional test statistic for ECS and adjust the test statistic. This improvement is definitely crucial to ECS. But in my view, this is a fix to a long standing issue of ECS rather than a new method in itself. Even with this improvement, the ECS is still not a completely statistically solid method because it still involves heuristic procedures to determine how much the ECS statistics should be adjusted. There is still no methodological guarantee of controlled type I error.

[Editors’ note: further revisions were suggested prior to acceptance, as described below.]

Thank you for resubmitting your work entitled "A conditional gene-based association framework integrating isoform-level eQTL data reveals new susceptibility genes for schizophrenia" for further consideration by *eLife*. Your revised article has been evaluated by Molly Przeworski (Senior Editor) and a Reviewing Editor.

The manuscript has been improved but there are some remaining issues that need to be addressed, as outlined below:

Reviewer #3 has 2 remaining issues that we would like you to more fully address before we would consider accepting the manuscript. Please read these comments carefully and revise accordingly. Thank you.

*Reviewer #1 (Recommendations for the authors):*

I would like to thank the authors for their detailed response to my questions and concerns. I have no further comments on the manuscript.

*Reviewer #2 (Recommendations for the authors):*

The authors have resolved all my concerns. Nice job.

*Reviewer #3 (Recommendations for the authors):*

The revised manuscript was submitted in PDF format with track changes. Although it clearly shows what's been changed since the original submission, it is therefore very difficult to read the complete article for a second overall assessment. I will therefore focus on comments I raised in my first review.

Two major issues remain.

1. The authors failed to respond properly to my comment on the overall concern that conditional analysis may not be the best approach for this type of problem. Figure 1 of SuSiE paper illustrates the exact scenarios why this could be a bad idea, which motivated my A-B-C genes example. Also incidentally, it is not true that SuSiE cannot work with summary statistics.

2. The response to the question of the choice of c (from another reviewer and myself) is still not satisfactory. It seems the favorable c still was computed in very specific scenarios. In fact, the new Figure 9 exactly demonstrates my concern: optimal choice of c depends greatly on LD and effect size, and for variants with small effects (OR < 1.4 which would be the case for most GWAS signals) the range of optimal c is wide. Given this result, I remain unconvinced that the tests are calibrated under current choice of c. From Figure 9 I would conclude that a conservative c = 1.8 should be used in general, which is different from the recommended choice from the authors.

---

## [Author Response]

Essential revisions:1) Systematic comparisons with existing methods are needed to demonstrate the potential gain in biological insight from this method. How do the results from MGCA_Dist, MGCA_eQTL, and MGCA_isoQTL compare with MAGMA? Can you include a direct comparison of MGCA with MAGMA, including the gene-based results, biological pathway enrichments, and drug repositioning analyses? Can the authors compare MGCA_eQTL to other eQTL-based approaches such as S-PrediXcan?

We thank the reviewers for their suggestions, and we agree that systematic comparisons with existing methods can help demonstrate the potential gain of our framework. Before the comparison with other methods, we recapped the inheritance and the extensions of our framework (eDESE) with our previous methods (i.e., ECS and DESE) in "Overview of the present study" of the revised manuscript, as suggested by Reviewer #3. We emphasized the two important advantages of the eDESE over ECS and DESE. We renamed MCGA_Dist as eDESE:dist and renamed MCGA_eQTL and MCGA_isoQTL as eDESE:gene and eDESE:isoform, respectively (see lines 110-139 in the revised manuscript). To clearly address all the reviewer's questions in point 1, we have organized these questions as follows and will address them in order:

Question A: How do the results from MGCA_Dist, MGCA_eQTL, and MGCA_isoQTL compare with MAGMA?

To discuss the limitations of MCGA in our last manuscript, we mentioned the number of significant genes identified by MAGMA to suggest that though the number of significant genes identified by MCGA (especially MCGA_isoform) was a little big, so was MAGMA. Considering that MCGA is a conditional gene-based association approach, we had no intention of directly comparing MCGA with MAGMA in our last manuscript.

Question B: Can you include a direct comparison of MGCA with MAGMA, including the gene-based results, biological pathway enrichments, and drug repositioning analyses?

Yes. Because conventional MAGMA and eDESE:dist both can perform gene-based association analysis by mapping variants to genes based on the variant-gene physical distance (+/-5kb around the gene boundary), we compared the susceptibility genes identified by eDESE:dist with that identified by MAGMA from the aspect of the significant gene counts, GO enrichment terms, text-mining results and the druggability of the susceptibility genes (see lines 208-245 and 434-478 in the revised manuscript).

Question C: Can the authors compare MGCA_eQTL to other eQTL-based approaches such as S-PrediXcan?

Yes. Since S-PrediXcan and eDESE (eDESE:gene and eDESE:isoform) both can perform the gene-based association analysis by considering the genetic regulatory effects on transcriptome (i.e., eQTLs), we compared the susceptibility genes identified by eDESE with S-PrediXcan.

The precalculated data in the PredictDB repository (http://predictdb.org) is important to S-PrediXcan. However, PredictDB provides gene-level expression and splicing predictors, but no isoform-level expression predictors, based on the MASHR models. Besides, there is no tutorial for the MASHR models in the PredictDB for now (see details in https://groups.google.com/g/predixcanmetaxcan/c/tTYlg8HIBeA/m/QNVskMCUCAAJ). So, we performed S-PrediXcan analysis only based on the gene-level expression predictors based on the MASHR models.

Before comparing S-PrediXcan and eDESE, we firstly optimized five brain regions based on eDESE:dist, eDESE:gene and eDESE:isoform (see lines 281-313 in the revised manuscript). Then, based on the five optimized brain regions, we compared the significant gene counts, GO enrichment terms, enriched consensus modules, text-mining results of the susceptibility genes (see lines 315-432 in the revised manuscript). We also compared the enrichment of drug-gene interactions and druggability of the susceptibility genes identified by MAGMA, S-PrediXcan and eDESE (see lines 434-478 in the revised manuscript).

The comparisons of eDESE with the two benchmark methods (i.e., MAGMA and S-PrediXcan) showed our framework's utility and potential gains.

2) When comparing MCGA-eQTL and MCGA-sQTL, only power is considered. The authors should include analyses to demonstrate the performance in control for false positives.

Thanks for pointing out this important question! In our last manuscript, we reported the power of isoform-level eQTLs over that of gene-level eQTLs in association analysis but neglected the performance in control for the false positives. In the revised manuscript, we reported the powers and false positives of gene/isoform-level eQTLs in association analysis. Our results showed that the false positives were controlled within a reasonable level (on average < 0.1% by a *p*-value cutoff of 0.001). Besides, the power of isoform-level eQTLs was higher than that of gene-level eQTLs in all scenarios in which gene expression can affect the phenotype (see lines 247-279 in the revised manuscript).

3) When choosing a favorable exponent value c (1.432 chosen in the study), the authors found that the c value is robust to trait type, sample size or variant size, but the authors didn't explain what factors affect the choosing of c. Considering the potential application of MCGA method in other studies, the authors should explain what factor affects c value, and provide the guidance how to choose an optimal c. Can the authors show via simulations the relationship between the coefficient used to adjust the chi square statistic correlations, $c$, and the strength of association signals?

Thanks for pointing out these important questions! To clearly address all the reviewer's questions, we have organized the questions as follows and will address them in order:

Question A: what factors affect the choosing of c?

Motived from the boundary of chi-square correlation, we adopted simulation studies to empirically choose *c* for controlling the type I error of the effective chi-square test. Besides the correlation of chi-square statistics, the choosing of *c* for the effective chi-square test may also be affected by the approximated non-negative solutions. However, the correlation of chi-square statistics is the major factor. Our simulation showed that the derived boundary and influence trend of LD on chi-square statistics were also applicable to the effective chi-square test. In the revised manuscript, we showed that the correlation of chi-square statistics is affected by the non-centrality parameter of chi-square statistics (see lines 640-655 in the revised manuscript).

Question B: how to choose an optimal c?

As the optimal *c* for controlling the type I error of the effective chi-square test would be affected by the non-centrality parameter of chi-square statistics which are generally unknown in practice, we have to resort to a grid search algorithm to explore an empirically optimal *c*. In our last manuscript, we mixed the methods of choosing optimal *c* with the introduction of new effective chi-squared statistics. We wrote a new subsection in Materials and methods to describe the procedure of choosing the optimal *c* in the revised manuscript (see lines 610-628 in the revised manuscript).

Question C: Can the authors show via simulations the relationship between the coefficient used to adjust the chi square statistic correlations, $c$, and the strength of association signals?

Yes. In the revised manuscript, we further investigated how the strength of association signals influenced the chi-square statistic correlations. Consistent with the theoretical derivation, the correlation of chi-square statistics quickly approached the maximal value |*r*| as the effect size increased. A stronger association signal led to a smaller *c,* and the *c* can quickly approach 1 as the association signals increased even in a moderate sample size (1000 cases and 1000 controls) (see Figure 9).

Reviewer #1 (Recommendations for the authors):– How do the results from MGCA_Dist, MGCA_eQTL, and MGCA_isoQTL compare with MAGMA? Can you include a direct comparison of MGCA with MAGMA, including the gene-based results, biological pathway enrichments, and drug repositioning analyses?

We thank Reviewer #1 overall for the numerous insightful and helpful suggestions and comments. Please refer to Question A and B of Essential Revisions point 1.

– Similarly, can you compare MGCA_eQTL to other eQTL-based approaches such as S-PrediXcan? I would like to see a systematic comparison with existing methods to see if novel biological insights can be derived from your results.

We thank the reviewer for this suggestion and comment. Please refer to Question C of Essential Revisions point 1.

– Does MCGA provide information on tissue-specific eQTLs/isoQTLs? This information is available in GTEx but is not reported for association results in supplementary tables. Tissue-specific effects for each tissue might improve the interpretation of results.

We thank the reviewer for this suggestion. In our last manuscript, we provided the gene-level and isoform-level eQTLs for each tissue but did not provide tissue-specific eQTLs/isoQTLs. Indeed, GTEx v8 provided the tissue-specific gene-level eQTLs and splicing QTLs for each tissue. However, we cannot directly access the original tissue-specific eQTLs because the download is not free.

Fortunately, we found a processed gene-level tissue-specific eQTLs set of GTEx v8 from the E-MAGMA tutorial document (https://github.com/eskederks/eMAGMA-tutorial), in which the tissue-specific SNP-gene associations were provided. We downloaded the E-MAGMA annotation files for Brain-Front Cortex (BA9) and Brain-Cerebellum, i.e., Brain_Frontal_Cortex_BA9.genes.annot and Brain_Cerebellum.genes.annot, from https://github.com/eskederks/eMAGMA-tutorial/blob/gtex_v8/emagma_annot_1.tar.gz.

We then extracted the tissue-specific SNP-gene associations from the above two annotation files and converted the tissue-specific SNP-gene associations to the eQTLs file format required by the eDESE:gene. We then perform the eDESE:gene analysis based on the tissue-specific eQTLs of Brain-Front Cortex (BA9) and Brain-Cerebellum, respectively. Our results showed that no tissue was significantly associated with schizophrenia by using the tissue-specific eQTLs of Brain-Front Cortex (BA9) (cutoff *p*-value < 0.05/50=1e-3, Author response image 1), while eight brain regions were predicted to be significantly associated with schizophrenia by using the tissue-specific eQTLs of Brain-Cerebellum (Author response image 1) .

**Author response image 1. sa2fig1:** The tissue importance generated by eDESE:gene based on the tissue-specific eQTLs. (a) Brain-Front Cortex (BA9); (b) Brain-Cerebellum. The red dotted lines denote the significant threshold.

Interestingly, though the tissue estimation based on the tissue-specific eQTLs of Brain-Front Cortex (BA9) was not satisfying, we found several brain regions ranked high among all the fifty tissues, and Brain-Front Cortex (BA9) ranked the second in the full tissue list (*p*-value = 2.4E-2). Brain-Cerebellum itself ranked second and was significantly associated with schizophrenia by using the tissue-specific eQTLs of Brain-Cerebellum. Further, we also investigated the potential susceptibility genes identified by eDESE:gene based on the tissue-specific eQTLs of Brain-Front Cortex (BA9) and Brain-Cerebellum, respectively. We found that these potential susceptibility genes were enriched with few GO terms, in which there was no biologically sensible GO term.

As suggested by the reviewer, our result showed that eDESE could benefit from the tissue-specific eQTLs, though the results of phenotype-associated tissue estimation based on the tissue-specific eQTLs were not as good as that based on the single tissue eQTLs in the current condition. This might result from the small sample size and strict threshold in eQTLs identification in GTEx v8. We believe that future genetic studies based on the increased sample sizes might alleviate this problem and increase the power of tissue-specific eQTLs. Integrating the tissue-specific eQTLs with eDESE will also be a topic for our further exploration. We also cited the paper and acknowledged the data in our Acknowledgments.

– Can you provide more information on WGCNA results? Are MCGA susceptibility genes enriched in a particular co-expression module? Are synaptic signalling related biological pathways enriched in a particular module? How did the network modules compare across brain tissues? Is it possible to build a consensus network for the top five brain regions (or indeed all brain regions in GTEx) to simplify the results?

We thank the reviewer for these suggestions and comments and for the opportunity to clarify.

In our last manuscript, we investigated the normalized intra-module connectivity score of the potential susceptibility genes identified by MCGA_eQTL and MCGA_isoQTL in each potential susceptibility brain region. The measurements were the statistics and statistical significance of the Wilcoxon rank-sum test based on the normalized intra-module connectivity score of the susceptibility and non-susceptibility genes in each tissue. In our last manuscript, we used Table 2 to present the tissues in which the intra-module connectivity score of the susceptibility was significantly higher than that of the non-susceptibility genes. We had no intention of comparing the network modules across brain regions.

We appreciate the reviewer's recommendation to build a consensus network to simplify the results, which we were previously unaware of. Following the reviewer's suggestion, we built a consensus network based on all thirteen brain regions' gene-level expression profiles (GTEx v8). We found that the potential susceptibility genes in the Brain-Cerebellum and Brain-Front Cortex (BA9) identified by eDESE:gene, and the potential susceptibility genes in the Brain-Cerebellum identified by eDESE:isoform were all significantly enriched with the same consensus module, which was significantly enriched with plenty of neuronal and synaptic signaling-related GO terms. However, no enriched consensus module was found for the susceptibility genes identified by S-PrediXcan (see lines 384-402 in the revised manuscript). The consensus analysis results supported the utility of our framework.

– The drug repositioning analysis could be improved by drawing on comprehensive drug-gene information from hetionet (https://doi.org/10.7554/eLife.26726.001). This resource might further refine some of your top associations by accounting for anatomies, biological processes, side-effects, and symptoms.

We thank the reviewer for this suggestion and appreciate the reviewer's recommendation to use Hetionet v1.0 to refine our top associations. We were mainly concerned about the potential susceptibility of the identified genes to schizophrenia and considered the drug repositioning evidence as a supplement. One reason is that some of our top associations contain many non-coding genes and pseudogenes that are not druggable. Another reason is that drug repositioning is complex and usually needs the support of multi-omics data. Thus we only investigated the drug-gene interaction terms and potential druggability of the susceptibility genes in our analysis.

We also studied Hetionet v1.0 and found that its function "Connectivity search" is interesting and useful in investigating the potential associations in genes, anatomies, biological processes, side-effects, symptoms and so on. We then used Hetionet v1.0 to support the top phenotype-gene associations for the susceptibility genes exclusively predicted by eDESE:isoform (see lines 349-354 in the revised manuscript).

Reviewer #2 (Recommendations for the authors):GTEx v8 data contains samples from multiple tissues, and the significant gene-trait association in some tissue may be non-causal driven by gene expression correlation with causal tissues. It should be interesting to come up a new method to remove those false positive.

This is a very good recommendation! We thank the reviewer for this comment, which has allowed us to improve the estimation results of phenotype-associated tissues. The issue about how to remove the false positive in the gene-trait association analysis is complex. We attempted to alleviate the issue by removing those false positives in the phenotype-associated tissues. The gene-trait association in more reliable phenotype-associated tissues might be more likely the truly susceptibility genes.

The significant tissues identified based on eDESE:dist may not all be phenotype-associated due to the high correlations between brain expression profiles. Based on the eDESE:dist, all thirteen brain regions were ranked top by statistical significance and predicted as the significant phenotype-associated tissues for schizophrenia, no matter based on gene-level or isoform-level (transcript-level) expression profiles (see lines 281-286 in the revised manuscript and supplementary file 1d). For schizophrenia, the Brain-Frontal Cortex is the most known potential susceptibility region, while what roles of other twelve brain regions play in the development of schizophrenia remains obscure.

Our framework, eDESE, inherits the framework of DESE and can also predict potential phenotype-associated tissues and susceptibility genes with the guide of *cis*-eQTLs. What eDESE over DESE is: First, eDESE is built based on a new ECS, with which the type I error could be controlled within a proper level. Second, eDESE expands the conditional gene-based association analysis of DESE by using different SNPs sets, i.e., physically nearby SNPs (DESE did), gene-level and isoform-level variant-gene eQTLs associations.

We then assumed that if a tissue (say *T_1_*) is a phenotype-associated tissue, potential susceptibility genes identified by eDESE with the eQTLs of *T_1_* will be more likely to be phenotype-associated genes and be selectively expressed in *T_1_* or similar tissues. Using the isoform-level eQTLs, based on a more precise resolution, we found that the analysis results may be more powerful, as demonstrated in the simulations in the revised manuscript (see lines 247-279 in the revised manuscript). Thus, we integrated the outputs of eDESE:dist, eDESE:gene and eDESE:isoform. Based on the principle of "the wisdom of the crowd", we finally optimized five brain regions, which were all predicted as the potential susceptibility tissues by eDESE:dist, eDESE:gene and eDESE:isoform, respectively.

This is also our first trial to remove those non-direct phenotype-associated tissues. The method might be rough but can be a reference for further explorations.

Reviewer #3 (Recommendations for the authors):My major concern is a lack of innovation as a new method. MCGA is based on DES, which integrates expression data with GWAS; and DESE relies on ECS to compute gene-level association evidence. Both DESE and ECS were developed by the authors in the past and published elsewhere. The only innovation in the MCGA methodology seems the improved control of type I error by formally assessing the null distribution of the conditional test statistic for ECS and adjust the test statistic. This improvement is definitely crucial to ECS. But in my view, this is a fix to a long standing issue of ECS rather than a new method in itself. Even with this improvement, the ECS is still not a completely statistically solid method because it still involves heuristic procedures to determine how much the ECS statistics should be adjusted. There is still no methodological guarantee of controlled type I error.

We understand the concern of Reviewer #3 about the novelty of our study. Indeed, MCGA (called eDESE in the revised manuscript) is based on the ECS and DESE, which were our previously published work. eDESE had at least two advantages over ECS and DESE. First, as the reviewer pointed out, we made a crucial improvement for the ECS to control the type I error by formally assessing the null distribution of the conditional gene-based association analysis based on the ECS and adjust the test statistic. Second, different from DESE, eDESE can perform conditional gene-based association analysis not only by mapping variants to genes according to their physical distance (like DESE), more importantly, but also based on gene-level and isoform-level gene-variant eQTLs associations. These improvements can help eDESE to use the distal variants to predict more significant susceptibility genes and tissues. We also found that isoform-level eQTLs could help eDESE identify more susceptibility genes than gene-level eQTLs in association analysis and produce the potentially significant phenotype-associated isoforms (or transcripts). Though mainly limited by sample size, the eQTL-guided models can be an important supplement for the gene-based approach. The methodology improvement of eDESE, the simulation studies and the comparisons with other methods demonstrated, at least partly, the potential gains of our framework.

[Editors’ note: further revisions were suggested prior to acceptance, as described below.]

Reviewer #3 (Recommendations for the authors):The revised manuscript was submitted in PDF format with track changes. Although it clearly shows what's been changed since the original submission, it is therefore very difficult to read the complete article for a second overall assessment. I will therefore focus on comments I raised in my first review.

We apologize for the inconvenience in assessing our revised manuscript in PDF format. To address this issue, we have provided the clean version of the revised manuscript through the "Related Manuscript File" option in the *eLife* submission system to facilitate the review process. We thank Reviewer #3 overall for the insightful and helpful comments and for the opportunity to clarify the confusion.

However two major issues remain.1. The authors failed to respond properly to my public comment on the overall concern that conditional analysis may not be the best approach for this type of problem. Figure 1 of SuSiE paper illustrates the exact scenarios why this could be a bad idea, which motivated my A-B-C genes example. Also incidentally, it is not true that SuSiE cannot work with summary statistics.

We thank the reviewer for these comments and for the opportunity to correct and clarify. Yes, SuSiE can work with summary statistics. We apologize for not adequately addressing the difference between SuSiE and our approach in our last response.

The role of our conditional association analysis is to remove redundant association signals per se. We agree with the reviewer that the conditional association analysis may not be the best approach for the fine-mapping analysis because the gene order of entering the conditional association analysis will influence the performance. Gene enters the conditional association analysis in a pre-defined order. In conventional conditional association analysis, the gene order can be determined according to the *p*-value of the gene's overall association with the phenotype. The conditional *p*-value of the second gene was obtained by conditioning on the first gene, and that of the third gene was obtained by conditioning on the first two genes. The conditional *p*-values of subsequent genes were calculated according to the same procedure. Thus, genes with priority to enter the conditional iteration tend to be significant, while genes entering the conditional iteration later tend to be insignificant. In the A-B-C gene example, as the reviewer pointed out, if the non-causal gene C has a stronger marginal signal than either true susceptibility gene A or B, then gene C will enter the conditional association analysis procedure first, which might lead to the true susceptibility genes (A and B) to be insignificant in the conventional conditional association analysis. Also similar to Figure 1b of SuSiE, conventional conditional association analysis may incorrectly identify non-causal gene C as the significant phenotype-associated gene, and gene A and B's association signal will become weaker by conditioning on gene C.

To address this issue, in this study, we proposed to use the gene selective expression (not the conventional association *p*-value) to guide the gene order of entering the conditional association analysis, which worked well in the real data analysis. The main assumption of gene selective expression is that the true susceptibility genes tend to selectively express in phenotype-associated tissues (e.g., Nat Genet. 2018;50:621–9. Genome Biol. 2019;20(1):233), which might result in a relatively high gene selective expression score for the true susceptibility genes and prioritize the true susceptibility genes to enter the conditional association analysis. For the A-B-C gene example, though gene C has a stronger marginal signal than either gene A or B, according to our assumption, the true susceptibility gene A or B might selectively express in phenotype-associated tissues and tend to have a higher gene selective expression score than gene C, thus can enter the conditional association analysis before gene C. Finally, by conditioning on the true susceptibility gene A or B, the non-causal gene C will be insignificant, which might help alleviate the problems in the A-B-C gene example.

In addition, as the reviewer pointed out, SuSiE is also a solid and efficient fine-mapping approach. Incorporating the advantages of SuSiE with our method will be interesting and helpful, which we will carefully consider in our future research. As SuSiE is an important fine-mapping tool, we also cited the original paper of SuSiE and discussed the relevant issue in the revised manuscript. We have added the following introduction about SuSiE in our revised manuscript (or see lines 62-65 of the clean version manuscript):

"SuSiE (sum of single effects), a novel and popular approach to variable selection in linear regression, can use summary statistics and LD to produce gene-level evidence of association in terms of Bayes Factor^10^."

We have also added the following discussion in our revised manuscript (see lines 562-566 of the clean version manuscript):

"Third, the performance of conditional association analysis for fine-mapping can be greatly influenced by the gene orders of entering the analysis. Though the improved effective chi-squared test with eDESE can work well in the real data of schizophrenia, integrating some non-conditional fine-mapping methods, such as SuSiE, with eDESE is worth trying."

2. The response to the question of the choice of c (from another reviewer and myself) is still not satisfactory. It seems the favorable c still was computed in very specific scenarios. In fact, the new Figure 9 exactly demonstrates my concern: optimal choice of c depends greatly on LD and effect size, and for variants with small effects (OR < 1.4 which would be the case for most GWAS signals) the range of optimal c is wide. Given this result, I remain unconvinced that the tests are calibrated under current choice of c. From Figure 9 I would conclude that a conservative c = 1.8 should be used in general, which is different from the recommended choice from the authors.

We thank the reviewer for these comments and for the opportunity to clarify. The *c* in Figure 9 was merely the correlation of chi-square statistics, which increased with OR up to the upper bound (|r|). We initially performed relevant analysis and drew Figure 9 to respond to the comment 3 of Reviewer #3 in our last response. The meaning of *c* value in Figure 9 is different from the *c* value used for the association test.

The factors that affect the choosing of favorable *c* for the association test are complex, including such as the correlation of chi-square statistics, the redundancy of degree of freedom and the approximated non-negative solutions. We generated the favorable *c* for the association test by extensive simulations. Specifically speaking, we simulated many relatively common scenarios, which included various sample sizes and variant numbers for both binary and continuous traits, respectively. The description of the common scenarios is as follows (or see lines 627-631 in the clean version manuscript).

"We considered in total 84 parameter settings, i.e., a combination of three different sample sizes (10,000, 20,000 and 40,000) and 14 different variant number (10, 30, 50, 80, 100, 125, 150, 200, 250, 300, 400, 500, 800, and 1000) for both binary and continuous traits, respectively. For a parameter setting, 40,000 datasets were simulated and used to produce *p*-values to determine the favorable *c* value."

We apologize for the confusion about the *c* in Figure 9. We have replaced the expression of *c* in Figure 9 with *c'* to distinguish it from the *c* for the association test (or see lines 640-665 in the clean version manuscript).

To further address the reviewer's concern, we also performed simulation analyses (similar to the analyses in Figure 4 in the revised manuscript) with *c*=1.8. The Q-Q plots in Author response image 2 simply that the *c*=1.8 led to a higher false-positive rate in the conditional association analysis. It seems that a larger *c* will make the test more liberal. Therefore, we chose a more conservative *c* (i.e., 1.432) in this study. We also added the following discussion in our revised manuscript:

"Fourth, the optimal *c* value of the effective chi-squared test is still empirical although we derived its range and relevant factors. The optimal *c* value can be improved to be better suited for other specific application scenarios."

**Author response image 2. sa2fig2:** Q-Q plots of the conditional, unconditional gene-based association test and likelihood-ratio test under the null hypothesis (*c*=1.8). (**a**) and (**d**), two gene-variant pairs with the similar variant number (SIPA1L2 with 29 variants and LOC729336 with 30 variants). (**b**) and (**e**): two gene-variant pairs with different variant numbers, and the first is larger than the second (CACHD1 with 41 variants and RAVER2 with eight variants). (**c**) and (**f**): two gene-variant pairs with different variant numbers, and the second is larger than the first (LOC647132 with five variants and FAM5C with 48 variants). (**a**), (**b**) and (**c**): the former gene has no QTL, and QTL explained 0.5% of heritability in the latter gene. (**d**), (**e**) and (**f**): the former gene has no QTL, and QTL explained 1% of heritability in the latter gene. Ten thousand phenotype datasets were simulated for each scenario. Unconditional Eff. Chi. (the red) represents unconditional association analysis at the former gene by the improved ECS. Conditional Eff. Chi (the blue) represents conditional association analysis at the former gene conditioning on the latter gene by the improved ECS. The likelihood ratio test (the yellow) was conducted based on the nested linear regression models.